# Rational design of flavivirus E protein vaccine optimizes immunogenicity and mitigates antibody dependent enhancement risk

Flaviviruses are a family of related viruses that cause substantial global morbidity and mortality. Vaccination against one flavivirus can sometimes exacerbate disease caused by related viruses through antibody-dependent enhancement (ADE) or interfere with the efficacy of subsequent vaccines. To address this challenge, we develop a vaccine strategy by introducing G5C/G102C mutations into the flavivirus envelope (E) glycoprotein. These mutations promote E dimerization through the formation of an inter-chain disulfide bond that conceals the immunodominant and ADE-prone fusion loop epitope (FLE). We validate this design on E proteins from multiple flaviviruses through biochemical, antigenic, and structural analyses. The resulting vaccine candidate, CC_FLE sE, derived from the Zika virus (ZIKV) and formulated with an advanced supramolecular adjuvant, provides significant protection in female mice challenged with ZIKV and prevents ADE caused by a related flavivirus, Dengue virus. In genetically modified mice expressing diverse human immunoglobulin loci, ZIKV CC_FLE sE induces robust neutralizing antibody responses targeting key ZIKV E protein epitopes, including the E-dimer−dependent epitope (EDE), indicating that ZIKV CC_FLE sE can elicit protective antibody responses within the human naïve B cell repertoire. Therefore, CC_FLE sE represents a promising strategy for developing flavivirus vaccines that minimize ADE risk while maintaining high protective efficacy.

Human pathogenic flaviviruses are a family of related viruses responsible for significant morbidity and mortality worldwide. They are transmitted primarily by mosquitoes, including yellow fever virus (YFV), dengue virus (DENV), Japanese encephalitis virus (JEV), West Nile virus (WNV), and Zika virus (ZIKV), as well as ticks such as the tick-borne encephalitis virus (TBEV)[1]. In 2016, ZIKV spread to the Americas and emerged as an infectious agent which caused a global health crisis[2,3]. ZIKV infection can cause Guillain−Barré syndrome in adults, and severe fetal neuro-malformations and fetal death during pregnancy[3]. ZIKV infection primarily occurs through mosquito bites,

but sexual and vertical (mother-to-fetus) transmissions have also contributed to the recent epidemic[2]. Despite significant efforts, there is currently no licensed ZIKV vaccine for disease prevention. However, FDA-approved live-attenuated and/or inactivated vaccines are available for YFV, JEV, TBEV, and DENV[4].

Flaviviruses possess an ~11 kb positive-sense RNA genome, which is translated into a single polyprotein. This polyprotein is cleaved into three structural proteins: capsid (C), pre-membrane/membrane (prM/M), and envelope (E), along with several non-structural proteins (NS)[5]. The E protein, which is the major target of neutralizing antibody (nAb)

✉ e-mail: epozhars@umd.edu; yuxingli@umd.edu

responses and a key focus of vaccine development, consists of three ectodomains (DI, DII, and DIII). The ZIKV E protein adopts a homo-dimeric conformation on mature virion, similar to other flaviviruses. Notable nAb epitopes on the ZIKV E protein include the quaternary E-dimer epitope (EDE)[6,7], the domain III lateral ridge (DIII LR) epitope[8,9], and other epitopes that have been identified at the dimer–dimer interface represented by mAb ZIKV-117[10,11] and elsewhere on the E protein[12–15].

There is significant sequence homology and structural similarity between the ZIKV E protein and those of other flaviviruses, such as DENV. An immunodominant but suboptimal epitope, the fusion loop epitope (FLE), spans 12 amino acid residues within DII and is highly conserved in flavivirus E proteins, eliciting abundant antibodies that demonstrate broad flavivirus cross-reactivity, but limited neutralization capacity[9,16,17]. Such cross-reactive antibody responses to flaviviruses, acquired through infection or immunization, can significantly affect the immune response and clinical outcomes of subsequent infections or immunizations. These effects may occur through mechanisms such as antibody-dependent enhancement of infection (ADE)[18] or immune imprinting[19,20].

ADE was first observed in DENV infections, where cross-reactive antibodies facilitate infection by different DENV serotypes via Fcγ receptor (FcγR)-mediated virus entry into myeloid cells[21–23]. Previous clinical studies on DENV vaccines have shown that DENV-naïve children, after vaccination, became more susceptible to severe dengue disease upon natural infection, likely due to ADE by vaccine-induced antibodies that bind but do not neutralize other DENV serotypes[24,25]. Moreover, animal studies[26–28] and clinical research[29] suggest that cross-reactive anti-ZIKV antibodies, primarily stimulated by ZIKV infection, may enhance the severity of subsequent DENV infections through ADE, mediated by cross-reactive antibodies.

Cross-reactive ADE-prone antibodies target conserved epitopes across flaviviruses, including those on the prM protein and the FLE of the E protein[30,31]. The prM and E proteins, expressed together as prM-E, are common antigens in current ZIKV vaccine candidates[2]. However, these vaccine candidates may elicit diverse antibody responses, including protective antibodies targeting nAb epitopes and weak or non-neutralizing antibodies, such as FLE-specific antibodies, that can cross-react with DENV E proteins and potentially exacerbate DENV infections[15,26], given that a majority of regions experiencing ZIKV epidemics are mostly endemic for DENV. Additionally, flavivirus cross-reactive antibody responses can reduce the effectiveness of vaccines for other flaviviruses due to immune imprinting[19,20]. For example, preexisting TBEV immunity has been shown to affect the quality of the immune response to subsequent YFV vaccination[32] and vice versa[33]. Furthermore, previous immunization with JEV or YFV vaccines can significantly inhibit the immune response to a subsequent inactivated ZIKV vaccine[34,35].

To ensure the safety and efficacy of future flavivirus vaccines, it is essential to address the potential risks associated with ADE and the inhibition of vaccine responses to heterologous flaviviruses and implement effective countermeasures. Several strategies have been proposed to mitigate the induction of ADE-prone antibodies, particularly those targeting the FLE on the viral E protein. One approach involves using ZIKV prM-E mutants with specific point mutations in the FLE to reduce its immunogenicity[36]. Another strategy includes engineering dimeric soluble E protein (sE) with FLE "stapled" to an adjacent DIII to minimize FLE exposure[37]. However, these modified immunogens have so far demonstrated suboptimal protection efficacy[36,37], as evidenced by significant viral loads observed in immune animals following ZIKV challenges.

In this study, we employ a structure-based approach to design flavivirus vaccine candidates with improved immunogenicity and reduced ADE potential. Our lead vaccine candidate, CC_FLE sE, which incorporates mutations to facilitate a disulfide bond linkage between the FLE and the N-terminus of the flavivirus E glycoprotein is characterized biochemically and antigenically for several flaviviruses including JEV, WNV, and ZIKV. These mutations result in an enhanced E-dimeric context and minimized FLE exposure. We have demonstrated the protective efficacy of the ZIKV CC_FLE sE design in mice, showing no ADE effect for DENV infection compared to the WT sE. Furthermore, we isolated potent monoclonal ZIKV nAbs from ZIKV CC_FLE sE-immunized human Ig loci transgenic mice that target major ZIKV E protein nAb determinants, including the quaternary EDE. To guide future vaccine development, we have also determined the structure of OZ-D4, a representative EDE nAb, in complex with the ZIKV sE dimer at atomic resolution and characterized its clonal affinity maturation.

Our CC_FLE design strategy could be instrumental in developing a range of flavivirus vaccines by reducing undesirable FLE-directed antibody responses while retaining protection efficacy.

## Results

### Structure-based vaccine design of ZIKV E proteins

Numerous studies suggest that the flavivirus FLE is the major ADE-prone epitope. On the mature virion, the ZIKV E protein is arranged as antiparallel homodimers and recognized by a class of quaternary epitope EDE-directed nAbs. Recombinant soluble forms of the ZIKV E protein ectodomain, known as sE, which lack the stem and anchor portions, are expressed predominantly as monomers[37]. These monomers bind poorly to EDE nAbs but show high affinity for FLE-directed, ADE-prone non-neutralizing antibodies[37]. At high in vitro concentrations, dimeric ZIKV sE can appreciably form[16,38]. The structure of the dimeric sE[16,38] suggests that in a closed dimeric context, the ZIKV FLE is heavily occluded by the DI, including the N-terminus, and DIII elements of the adjacent E protein protomer[39,40] (Fig. 1A, upper left panel). In contrast, the FLE is readily exposed in the monomeric context or in dynamically opened E dimers (Fig. 1A, lower left panel), a phenomenon known as "viral breathing"[39]. The accessible surface area (ASA) of the entire FLE in the dimer context is ~459 Å² (38 Å²/residue) (Fig. 1A, right). In contrast, in the monomer context, the FLE shows a substantially increased exposure, with an ASA of 836 Å² (70 Å²/residue), representing an >80% increase (Fig. 1A, right). Among the 12 amino acid residues of FLE, three residues - D98, W101, and F108- exhibit ASA values increased by >50% (Fig. 1A, right) due to the dimer-monomer transition, consistent with the observation that residues W101 and F108 are major contact points for FLE-specific ADE-prone antibodies[16,36].

Conceptually, the monomeric form of sE is not an ideal immunogen for stimulating protective immune response due to: (i) the poor presentation of quaternary EDE nAb epitopes, which are more readily formed in a dimeric context (Fig. 1A), and (ii) the extensive exposure of weak/non-neutralizing FLE epitope, which could robustly elicit ADE-prone antibodies (Fig. 1A). Therefore, we first sought to engineer stable ZIKV sE dimer vaccine candidates to mimic the native E protein conformation by introducing one or two inter-protomer disulfide bonds in sE, based on structural modeling of sE (PDB: 5JHM) (Fig. 1B). The corresponding recombinant E protein constructs were designated as CC_FLE sE carrying mutations G5C/G102C, and CC_Core sE bearing mutation A264C (Fig. 1B, C) that has been described by previous studies to promote sE dimerization[37,41–43], respectively. Of note, the mutations in CC_FLE located within the FLE could further reinforce the occlusion of major FLE residues such as W101 and F108, thus dampening the antigenicity of FLE by tethering the FLE on one protomer to the N-terminus of the other protomer (Fig. 1B) and reducing the ADE effect.

Additionally, we adopted an alternative strategy to prevent potential ADE antibody elicitation by deleting DII (residues 51–133, and 195–284) or DI & DII (residues 1–302), which contain more conserved amino acid residues to form ADE-prone epitopes. This approach resulted in the constructs DI-DIII and DIII, respectively (Fig. 1C). To

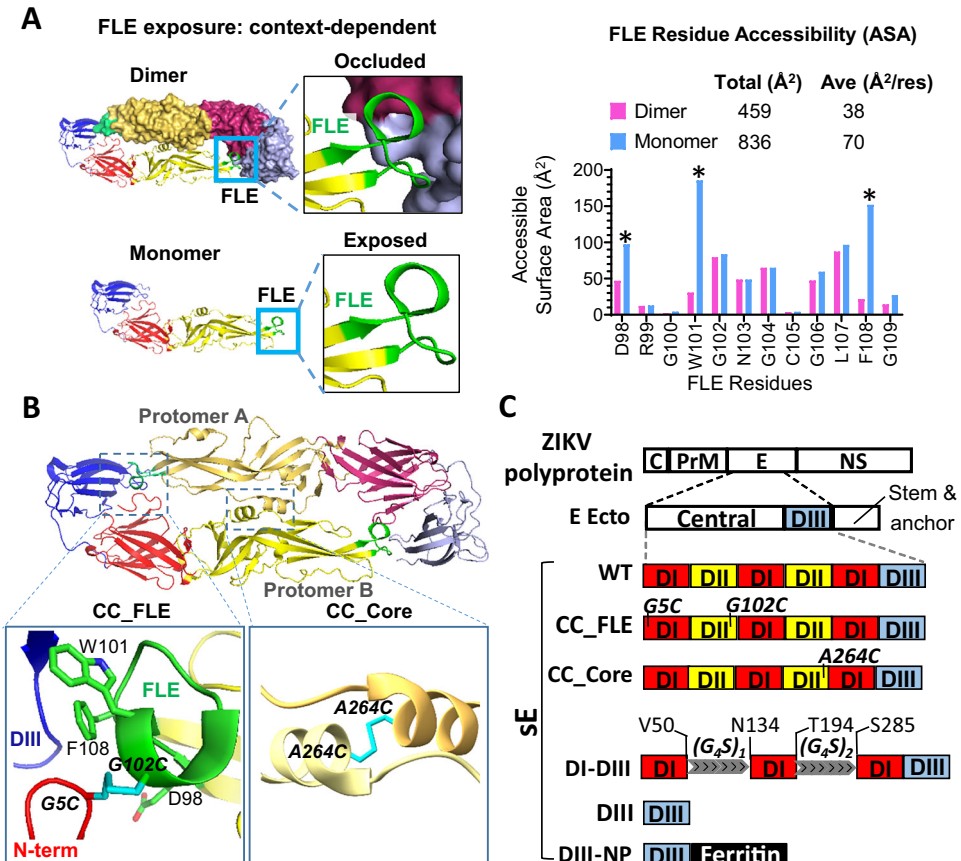

**Fig. 1 | ZIKV E-based immunogen design. A** Dynamic accessibility of FLE in ZIKV E protein (PDB: 5JHM). (*upper left*) FLE (green) on DII (yellow) is occluded by surrounding DI (red) and DIII (light blue) elements in a closed E dimeric context on the mature virion surface at high frequency, but (*lower left*) readily accessible in an E monomeric context or on an opened E dimer; (*right*) Accessible surface area (ASA) of each FLE residue and the whole FLE in ZIKV E assessed with PDBePISA is higher in the monomer context than the dimer context. Residues with >50% of this ASA change are denoted with an asterisk. Ave (Å²/res), average ASA of each FLE residue. **B** Structure-based design strategy to increase dimer formation propensity of ZIKV E by introducing cysteine mutations to form an inter-protomer disulfide bond (CC) linkage (modeled, in cyan) resulting in immunogens: (left) at the fusion loop epitope (FLE, green), designated as CC_FLE, containing mutations G102C (protomer A) & G5C (protomer B) to form a CC linkage tethering the FLE to the N-terminus of E (DI, red) of the neighboring protomer. Side chains of D98, W101, & F108, the FLE residues with more than 50% of exposure (ASA) reduction from monomer to dimer context, are denoted. (right) At the center of the E dimer interface, designated as CC_Core (mutation A264C). **C** Top, Schematic of the ZIKV E protein within the context of the viral polyprotein. The localization of the residues that form inter-protomer disulfide bonds upon mutation to cysteine is denoted. D, domain; C, capsid protein; prM, precursor membrane protein; NS, non-structural protein; sE, recombinant soluble ecto portion of ZIKV E protein (residues 1–405); WT, nascent sE; CC_FLE (G5C & G102C); CC_Core (A264C); DI-DIII, sE residues 51–133 and 195–284 are deleted and replaced with a $(G_4S)_1$ and a $(G_4S)_2$ linker, respectively; DIII, consists of residues 303–405; DIII-NP, residues 303-406 and ferritin motif connected by a peptide linker.

enhance the immunogenicity of DIII, we fused its C-terminus with the N-terminus of a gene fragment encoding a self-assembling virus-like nanoparticle (NP), ferritin[44,45], creating the vaccine candidate construct DIII-NP (Fig. 1C).

## Antigenicity, thermostability, and structure of rationally designed ZIKV E-based immunogens

We cloned the genes encoding ZIKV E-based immunogens listed in Fig. 1C into an insect expression vector and transformed *Drosophila* S2 cells for protein expression. The dimer-prone mutant sE proteins were purified by His-tag column followed by size exclusion chromatography (SEC) (Fig. S1A) to remove aggregates and monomeric proteins that failed to form inter-protomer disulfide bond. These two purification steps resulted in dimeric CC_FLE and CC_Core sE proteins with high homogeneity (Figs. 2A, and S1B), as expected. Differential scanning calorimetry profiles demonstrate substantially improved thermostability of CC_FLE sE compared to WT sE, with the thermal transition temperature (Tm) increased by 15 °C (from 43.5 °C to 58.7 °C) (Fig. 2B). In contrast, CC_Core sE showed a moderate (3 °C) increase in Tm compared to WT sE (Fig. 2B). To confirm the dimeric configuration of

CC_FLE sE, we determined the cryogenic-electron microscopy (cryo-EM) structure of the complex of CC_FLE sE and a ZIKV neutralizing antibody SMZAb2[46] at resolution of 4.09 Å, enabling detailed structural characterization. The resulting structure (Figs. 2C and S2A–G, and Table S1) revealed a symmetric CC_FLE dimer, with overall structure identical to previously published ZIKV sE dimer (Table S2, and Fig. S2L), engaged by two SMZAb2 Fab molecules. The 150-loop (residues 147–161)[47] of the E dimer lacks sufficient electron density, indicating that this region is disordered, similar to the wild-type ZIKV E dimer (PDB: 5LBS), and it does not appear to engage in direct contact with SMZAb2 in the structure. The Fabs bind across the dimer interface, contacting both sE protomers and targeting a quaternary epitope that spans domains DII and DIII of the E protein. Epitope analysis reveals that the SMZAb2 epitope significantly overlaps with that of EDE1-C8, a well-characterized broadly neutralizing antibody (Fig. S2I–K). Of the 32 residues engaged by SMZAb2, 27 are shared with the EDE1-C8 epitope (Fig. S2I), underscoring SMZAb2's classification within the EDE1 antibody family. The presence of the engineered disulfide bond (C5–C102) linking the protomers is clearly resolved, confirming successful covalent stabilization of the dimer interface

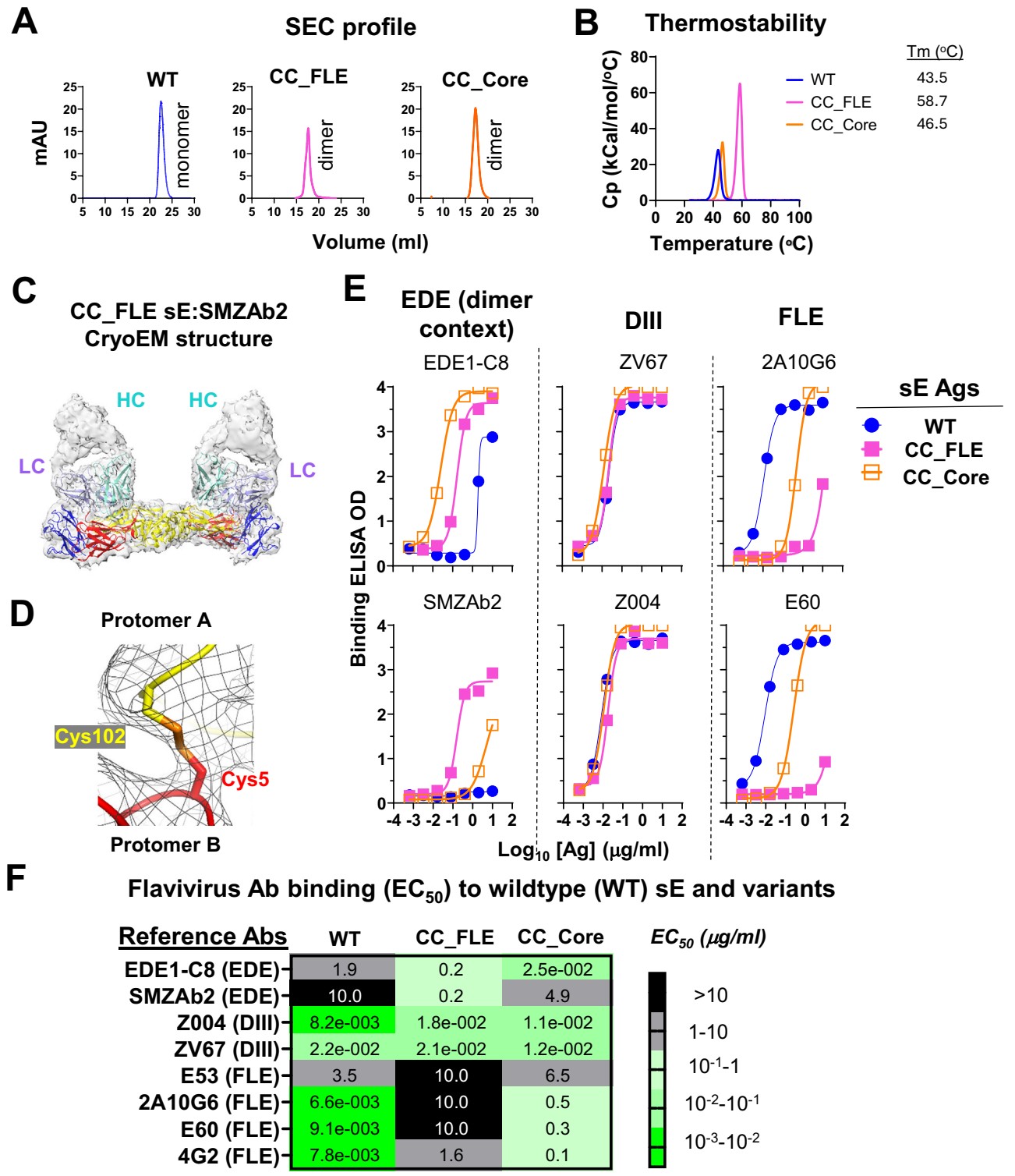

**Fig. 2 | Biochemical and antigenicity characterization of ZIKV sE mutants CC_FLE & CC_Core. A** Size exclusion chromatography profiles of the purified ZIKV sE mutants. **B** Differential scanning calorimetry profiles demonstrate improved thermostability of CC_FLE (magenta) and CC_Core (gold) over the WT sE (blue). **C** Overall structure of CC_FLE sE: SMZAb2 complex fitted into the semitransparent Cryo-EM density shown in grey. ZIKV E dimer is colored by domains: domain I (red), domain II (yellow), domain III (blue), fusion loop (green). SMZAb2 heavy chain (cyan), light chain (magenta). **D** Close-up view detailing the C5-C102 disulfide linkage (gold) connecting two protomers (yellow and red). **E** ELISA binding to reference ZIKV mAbs, including EDE epitope mAbs (EDE1-C8, SMZAb2), DIII LR (ZV67, Z004), and FLE (2A10G6, E60); **F** ELISA binding $EC_{50}$ value heat map. Each sample test in (**E**) and (**F**) was duplicated, and each assay in (**B**), (**E**), & (**F**) was repeated at least two times. Source data are provided as a Source Data file.

(Fig. 2D). As expected, exposure of the FLE loop, including residues W101 and F108, is markedly reduced due to occlusion by the adjacent E protomer in the dimer context (Fig. S2J, M).

To characterize the antigenicity of our immunogens, we assessed their binding affinity for a panel of reference mAbs, in comparison with WT sE. Consistent with the design rationale, WT sE, primarily in monomeric form, binds EDE quaternary epitope mAbs poorly. CC_FLE binds both EDE mAbs, EDE1-C8 and SMZAb2 well, while CC_Core only binds the former mAb well but the latter poorly (Fig. 2E). The DIII LR epitopes are well retained in both CC_FLE and CC_Core, as they bind DIII LR epitope mAbs, ZV67 and Z004, similarly to WT (Fig. 2E). Lastly, as expected, CC_FLE has nearly abolished binding to FLE mAbs 2A10G6 and E60 (Fig. 2E), while CC_Core shows significantly diminished binding compared to WT. We assessed the reactivity of more flavivirus FLE cross-reactive but weak or non-neutralizing mAbs and consistently noted that our dimeric CC_FLE sE, which carries an engineered inter-protomer disulfide bond locking the potential ADE epitope FLE to the N-terminus of the adjacent sE protomer (i) remarkably show enhanced EDE nAb epitope presentation compared to WT sE; (ii) well retain nAb epitopes on domain III; and (iii) all substantially dampen the display of FLE epitopes (Fig. 2F). Of note, the G5C and G102C mutations alone are not sufficient to abolish FLE–mAb binding, as a minor monomeric fraction of ZIKV CC_FLE sE still binds the FLE–mAb E60 (Fig. S1C), similar to WT sE. This suggests that the reduced FLE–mAb binding of CC_FLE sE results from occlusion of the FLE caused by the inter-chain disulfide linkage formed by the G102C/G5C mutations. As expected, DI-DIII and DIII immunogens bind a panel of DIII-specific nAbs, including ZV67 and Z004 (Fig. S1D–H) in a manner similar to WT sE protein, indicating well-retained antigenicity, with abolished binding to FLE mAb, 4G2 (Fig. S1D–H).

## CC_FLE linkage can potentially be applied to other flavivirus sEs to prevent ADE

Sequence homology analysis of the N-terminus and FLE regions of the sEs from various mosquito-borne flaviviruses revealed that residues G5 and G102, which were mutated to cysteine (C) in the ZIKV CC_FLE design to form the N-term/FLE disulfide linkage, are conserved in the corresponding sEs of these viruses (Fig. 3A). The potential applicability of this disulfide bond linkage in the sEs of other flaviviruses, such as JEV and WNV, was further supported by structural analysis of their envelope proteins (PDB: 3P54 & 3IYW). This analysis predicted a high likelihood for the formation of this disulfide bond. We then expressed and characterized the CC_FLE version of JEV and WNV sE, respectively, which showed a dimeric configuration after purification (Fig. 3B), in contrast to the predominantly monomeric JEV or WNV WT sE (Fig. 3B). Cryo-EM analysis of JEV and WNV CC_FLE sE proteins confirmed their dimeric configurations (Fig. 3C, Tables S1 and S2, and Fig. S3). We were able to further visualize the disulfide bond linkage facilitated by mutations G5C/G102C in JEV CC_FLE sE (Fig. 3D). As expected, in the JEV CC_FLE sE dimer, the ADE-prone epitope FLE is highly occluded by DI and DII from the adjacent sE protomer (Fig. 3E), similar to its occlusion in the sE dimers displayed on JEV virions[48]. Consistently, JEV and WNV CC_FLE sE displayed 7- to 300-fold decreased binding to FLE-specific mAbs, in comparison with the WT sE (Fig. 3F, G). Therefore, the CC_FLE design strategy could potentially be applied to prevent ADE of sE-based vaccines of other flaviviruses.

## CC_FLE elicits potent nAb responses in immune mice

We set to test the immunogenicity of the engineered ZIKV vaccine candidates described above including the CC-FLE sE (Fig. 1C), formulated with Adjuplex, a lecithin-based adjuvant[49] in C57BL/6 mice (Fig. 4A). After two immunizations on days 0 and 28, respectively, sera from mice immunized with CC_FLE showed geometric mean of $ID_{50}$ titers (~$10^4$) against ZIKV reporter virus particle (RVP) of H/PF/2013 strain, similar to WT sE (Fig. 4B). Sera from mice immunized with DIII or

DIII-NP/DIII (priming with DIII-NP/boost with DIII) showed tenfold lower $ID_{50}$ titers than WT sE (Fig. 4B), while immune sera resulted from DI-DIII immunization showed no neutralization activity (Fig. 4B), suggesting that the deletion of DI and/or DII of sE leads to attenuated immunogenicity. This is possibly due to protein misfolding or reduced stability resulting from the deletion. Surprisingly, the CC_Core dimer has elicited low nAb titers nearly equivalent to PBS inoculated mice group (Fig. 4B) after two immunizations, indicating mutation A264C impedes immunogenicity. Nevertheless, the observed robust nAb response from the immune mice inoculated with ZIKV CC_FLE dimer indicated premise for further development.

## Supramolecular polyphosphazene-based adjuvant formulation potentiates vaccine efficacy in mice

Polyphosphazene (*PPZ*) immunoadjuvants (Fig. S4) are well-defined water-soluble macromolecules with phosphorus-nitrogen backbone and organic side groups, which have been proven to dramatically potentiate immune responses to diverse viral and bacterial vaccine antigens[50,51]. The unique feature of these synthetic macromolecules is that, upon mixing in aqueous solutions, they are capable of self-assembling with vaccine antigens into nano-scale virus-mimicking complexes. The first-generation PPZ adjuvant, poly[-di(carboxylatophenoxy)phosphazene] (PCPP), has shown a dose-dependent immunopotentiation effect and excellent safety profile in five clinical trials[51]. Its structural homologue, poly[-di(carboxylatoethylphenoxy)phosphazene] (PCEP), demonstrated further ability to both enhance and modulate quality of the immune response[52,53]. A complementary immunoadjuvant, small molecule R848 (resiquimod), which is a clinical-grade TLR7/8 agonist, activates immune responses in TLR7/8 dependent mechanism and induces superior cytokine secretion, macrophage activation, and enhancement of cellular immunity[54–56]. Both PPZ and R848 technologies have been recently synergized by co-assembling antigen-PCPP complexes with R848, which resulted in non-covalently assembled supramolecular constructs that mimic virus-like features desirable for stimulating potent immune response: (i) 60–80 nm diameter, (ii) repetitive/multiple antigen display, and (iii) decoration with danger signals recognized by pattern recognition receptors[57].

Here, we attempted to test if PPZ in conjunction with R848 could better potentiate the immunogenicity of sE protein than Adjuplex, the adjuvant we used earlier. We demonstrated the formation of nano-scale ternary CC_FLE-PCEP-R848 complex in a unimodal size distribution with a z-average hydrodynamic diameter of 60 nm (Fig. S4) and the ability of complexed antigen to bind antibodies (Fig. S5) using Asymmetric Flow Field-Flow Fractionation (AF4) and dynamic light scattering (DLS) methods. We then focused on investigating the immunogenicity of our lead immunogen CC_FLE in various PPZ formulations including (i) PCEP, (ii) PCEP-R848, (iii) PCPP, (iv) PCPP-R848, (v) R848, and (vi) Adjuplex as control. We also added WT sE formulated with Adjuplex as a reference group, in the immunization study depicted in Fig. 5A. 1 week after the second immunization, sera from mice immunized with WT sE formulated with Adjuplex showed neutralization $ID_{50}$ titers against ZIKV H/PF/2013 RVP around $10^4$, similar to mice immunized with CC_FLE formulated with Adjuplex, PCEP, or PCEP-R848 (Fig. 5B), while sera from immune mice inoculated with CC_FLE formulated with PCPP or PCPP-R848 displayed 20–30 fold lower $ID_{50}$ titers (Fig. 5B). No neutralization activity was detectable in sera from mice immunized with CC_FLE formulated with R848, or CC_FLE alone (no adjuvant).

In addition to ZIKV RVP neutralization assay, we determined the neutralization capacity of selected sera from the immune mice group against an authentic virus, ZIKV FSS13025, by PRNT assay. Consistent with the RVP neutralization data, sera from mice immunized with CC_FLE formulated in either PCEP-R848 or Adjuplex all displayed potent neutralization capacity against ZIKV FSS13025, similar to sera

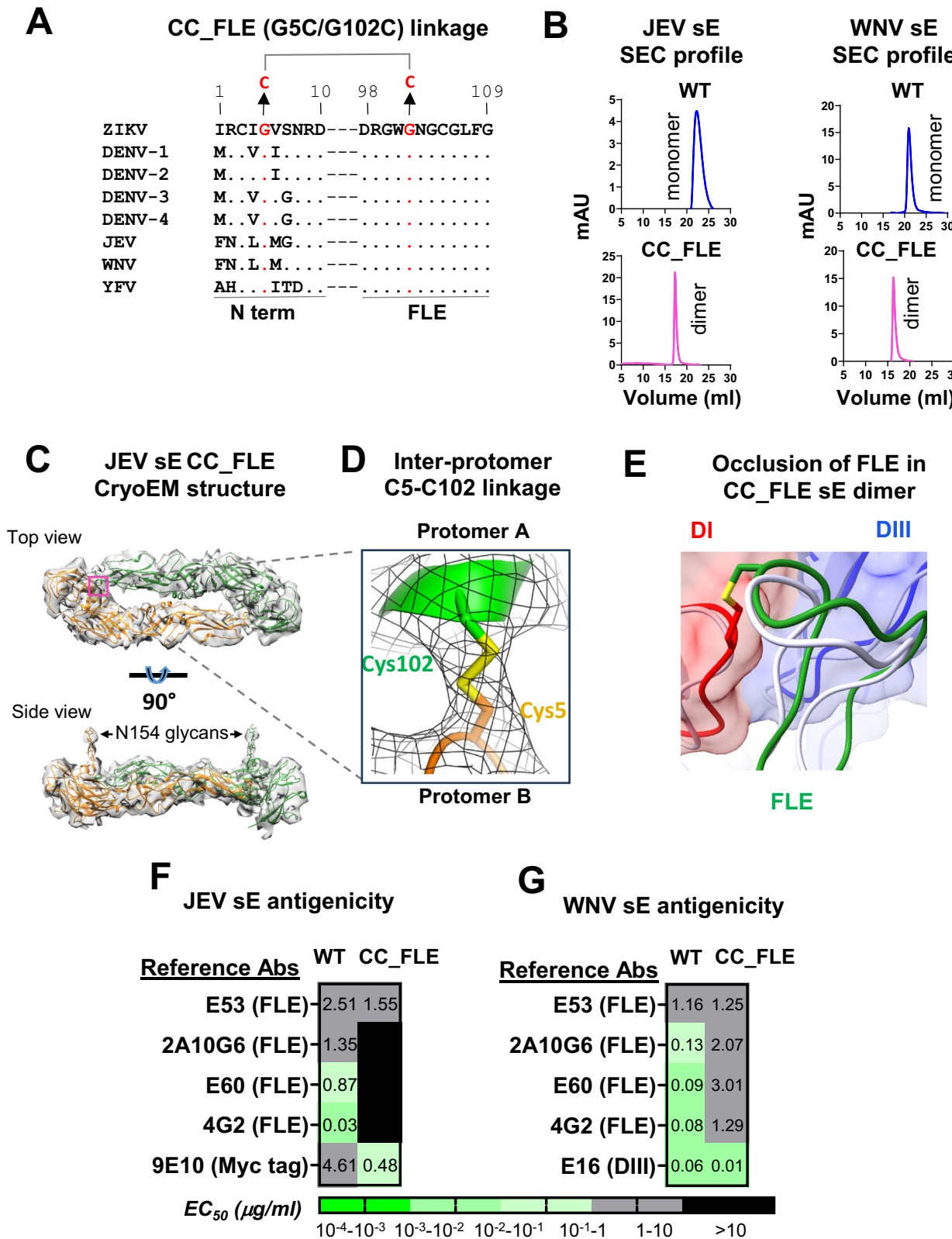

**A** CC_FLE (G5C/G102C) linkage

**B** JEV sE SEC profile / WNV sE SEC profile

**C** JEV sE CC_FLE CryoEM structure

**D** Inter-protomer C5-C102 linkage

**E** Occlusion of FLE in CC_FLE sE dimer

**F** JEV sE antigenicity

**G** WNV sE antigenicity

from mice immunized with WT sE formulated in Adjuplex (Fig. 5C). We then sought to determine protective efficacy of these selected potent immune sera in a passive serum transfer study (Fig. 5D). AG129 mice (IFNα/β/γR−/−), immunocompromised and highly susceptible to ZIKV infection, were infused with donor C57BL/6 mouse immune sera of day 42 (Fig. 5D), challenged with ZIKV FSS13025 1-h post serum infusion, and were monitored for 20 days post infection (DPI). AG129 mice

receiving sera from immune mice inoculated with CC_FLE formulated with PCEP+R848 adjuvant displayed significantly longer survival time (median, 15 days) upon challenge than from animals inoculated with PBS (median, 11 days) ($^*p < 0.05$, Mantel−Cox log-rank test) or WT sE formulated with Adjuplex (median, 11 days) (Fig. 5D). AG129 mice receiving sera from donor mice immunized with CC_FLE formulated with Adjuplex also showed median survival time of 13 days (Fig. 5D),

**Fig. 3 | Biochemical and antigenicity characterization of JEV and WNV CC_FLE sE mutants. A** Amino acid sequence alignment of the sE N terminus (N-term) and FLE of selected mosquito-born flaviviruses. Residues identical to ZIKV sE are indicated by dots. G5 and G102, mutated to cysteine (C) in the CC_FLE design to form a disulfide linkage, are highlighted in red. Abbreviations: DENV, Dengue Virus; WNV, West Nile Virus; JEV, Japanese Encephalitis Virus; YFV, Yellow Fever Virus. **B** Size exclusion chromatography profiles of JEV and WNV sE proteins; **C** Structural characterization of the JEV CC_FLE sE mutant. The overall dimer model is fitted into the Cryo-EM map, with two individual protomers colored in green and gold, respectively. *Upper:* top view with the interface between the FLE and the N-terminus highlighted in a magenta box. *Lower:* side view, with N154 glycans indicated; **D** Close-up view detailing the C5-C102 disulfide linkage (yellow) connecting two protomers (green and gold). **E** Occlusion of the FLE (green) by the surrounding DI (red) and DIII (blue) domains of the neighboring protomer in the JEV CC_FLE sE dimer, with the C5–C102 disulfide linkage shown in yellow. The superposed FLE loop from the JEV sE dimer (PDB: 5WSN) is shown in light gray. **F** ELISA binding $EC_{50}$ value heat map showing attenuated binding to FLE-specific mAbs for JEV CC_FLE sE compared to the WT. **G** Similar ELISA binding $EC_{50}$ value heat map for WNV CC_FLE sE compared to the WT. Each sample test was duplicated, and each assay was repeated at least two times. Source data are provided as a Source Data file.

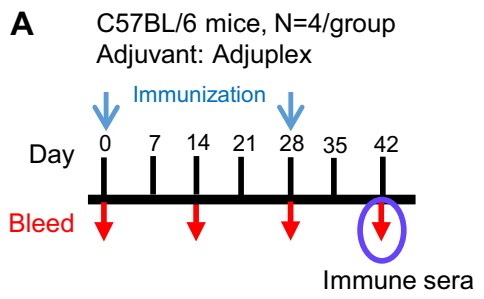

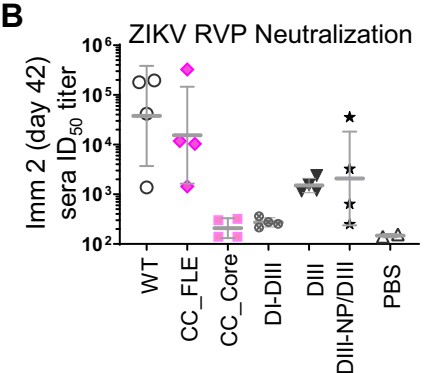

**Fig. 4 | ZIKV sE immunogen screening in C57BL/6 mice. A** Immunization schedule. C57BL/6 mice (*N* = 4/group) were immunized with 20 μg of immunogen formulated in Adjuplex adjuvant via subcutaneous route on days 0 and 28. Sera were collected on day 42 and tested for neutralization capacity. DIII-NP/DIII group mice were primed with DIII-NP on day 0 and boosted with DIII on day 28. **B** Neutralization ID50 titers of day 42 sera (*N* = 4/group) against RVP of ZIKV H/PF/2013. Shown are the ID50 titers of individual serum samples, with geometric mean ± standard deviation. Each sample test was duplicated, and each assay was repeated at least two times. Source data are provided as a Source Data file.

longer than PBS- or WT sE + Adjuplex inoculated group, although this is not statistically significant. AG129 mice receiving immune sera from CC_FLE and PCEP-R848 adjuvant-immunized animals also showed less body weight loss (Fig. S6A, statistically not significant) and lower health score (multiple *t* test, ***$p < 0.001$, days 11 and 12 post infection, Fig. S6B) than from PBS inoculated animals. Although the nAb titers of immune sera from animals inoculated with WT sE + Adjuplex, CC_FLE sE + Adjuplex, or CC_FLE sE +PCEP-R848 were similar, the better protection efficacy conferred by the sera from the mice immunized with CC_FLE sE +PCEP-R848 indicated that other protection mechanisms, such as antibody effector functions[58] including antibody dependent cytotoxicity (ADCC) may play a role in this AG129 mouse model.

In summary, formulation of our immunogen, CC_FLE into nano-scale ternary complexes with PCEP-R848 is beneficial for potentiating immunogen protection efficacy, indicated by (i) potent nAb titers of immune sera (Fig. 5B, C) and (ii) in vivo protection for ZIKV-challenged AG129 mice conferred by passive immune sera transfer (Fig. 5D).

## CC_FLE mutation largely abolishes ADE potential for enhancing DENV infection
To further evaluate the potential of CC_FLE inducing cross-reactive ADE-prone Abs, we incubated DENV serotype 1 (DENV-1) or serotype 2 (DENV-2) reporter virus particle with sera from ZIKV CC_FLE or WT sE immune mice (Fig. 5A), followed by co-incubation with K562 cells ($_{hu}$FcγRIIA⁺) to mimic Fc receptor-mediated ADE for DENV infection. As expected, most sera from mice immunized with ZIKV WT sE formulated with Adjuplex showed peak enhancement titer (PET) for DENV-1 or DENV-2 around 200 (Fig. 5E, F), and enhanced magnitude of infectivity for DENV-1 or DENV-2 above 200-fold (Fig. 5E, F) at PET. In contrast, most sera from mice immunized with ZIKV CC-FLE formulated with Adjuplex or PCEP-based adjuvant showed no significant enhancement for DENV-1 or DENV-2 infection (Fig. 5E, F) consistent

with the design rationale of CC_FLE which aims at abolishing the elicitation of ADE-prone antibody responses. The in vitro ADE data were confirmed by an in vivo study in which the enhancing effects of ZIKV immune serum on DENV-2 infection in AG129 mice were examined (Fig. 5G). Recipient AG129 mice were passively transferred with pooled donor serum from C57BL/6 mice vaccinated with ZIKV WT or CC_FLE sE formulated in Adjuplex (Fig. 5E, F), followed by challenge with a partial-lethal dose of DENV-2 (strain DS210). We found that AG129 mice receiving WT immune sera showed more severe mortality (**$p < 0.01$, Fig. 5G, left) and higher DENV-2 virus load (*$p < 0.05$, Fig. 5G, right) than PBS-treated mice, indicating a WT immune sera-mediated ADE effect. In contrast, the mice receiving CC_FLE immune sera showed a survival curve (Fig. 5G, left) and terminal tissue (lymph node) DENV-2 virus load (Fig. 5G, right) similar to those receiving PBS buffer, suggesting no ADE effect from the CC_FLE immune sera.

## ZIKV dimeric CC_FLE sE elicits nAb responses with distinctive quality compared to monomeric WT sE
To map the specificity of the nAb response elicited by ZIKV CC_FLE sE, we performed immune serum antibody depletion followed by ZIKV RVP neutralization assay. In this assay, pooled day 42 sera (Fig. 5A) from mice immunized with WT or CC_FLE sE formulated with Adjuplex were pre-mixed with sE protein variant antigens to deplete cognate serum neutralizing antibodies prior to ZIKV RVP incubation (Fig. S7). mAbs ZV67 (DIII-specific) and EDE1-C8 (quaternary epitope-specific) were used as controls (Fig. S7).

We found that the neutralization activity of immune sera from mice immunized with WT sE was reduced to below 50% RVP inhibition after depletion with all sE antigens, including WT, CC_FLE and DIII (Fig. S7), which all contain epitopes on DIII, similar to the DIII-specific mAb, ZV67 (Fig. S7). This suggests that the nAb responses elicited by WT sE are DIII-directed.

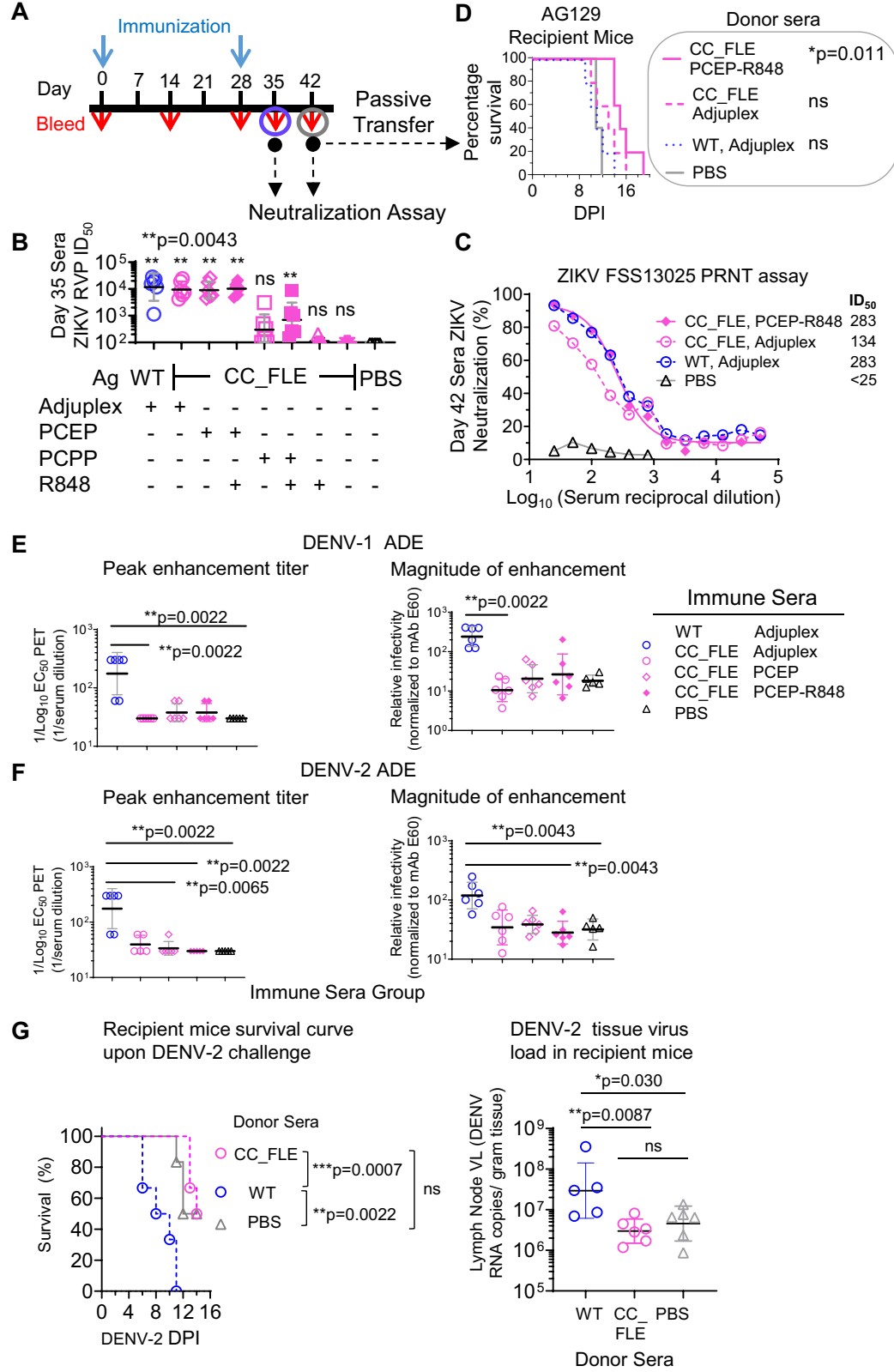

In contrast, immune sera from mice immunized with CC_FLE showed a distinct specificity profile: only dimeric CC_FLE could efficiently deplete the serum nAb activity (Fig. S7), similar to the quaternary epitope-specific mAb, EDE1-C8 (Fig. S7). Depletion with WT or DIII sE antigen only partially reduced serum neutralization capacity (Fig. S7), indicating additional DIII-directed nAb specificity.

Therefore, dimeric CC_FLE sE elicits multi-layers of nAb specificities, including those directed at quaternary epitopes and DIII, which are more sophisticated than the mainly DIII-focused nAb responses stimulated by WT sE. This suggests that the superior quality of nAb responses elicited by dimeric CC_FLE over WT sE immunogen may translate to enhanced protection efficacy.

**Fig. 5 | Optimization of adjuvants for ZIKV sE immunogen and elimination of ADE effects. A** Immunization schedule. C57BL/6 mice (*n* = 6/group) were immunized subcutaneously with 20 µg of immunogen formulated with different adjuvants. Sera collected on days 35 and 42 were analyzed by ZIKV neutralization assays (RVP and PRNT). Day 42 sera were also used for passive transfer studies. **B** ZIKV H/PF/2013 RVP neutralization titers (ID$_{50}$) from day 35 sera (*n* = 6/group). Data represent geometric mean ± geometric SD. Each sample was tested in duplicate, and assays were repeated at least twice. Statistical comparison between PBS and CC_FLE-immunized groups used two-tailed Mann–Whitney U-test. **C** Day 42 sera were tested against live ZIKV FSS13025 by PRNT. Each data point represents the mean of duplicate tests ± SD. **D** Passive protection study. AG129 mice (*n* = 6/group) received 200 µl of day 42 immune sera (from **A**) via intraperitoneal route and were challenged 1 h later with 10$^4$ PFU of ZIKV FSS13025. Survival was monitored post-

challenge (DPI, days post infection). Mantel−Cox log-rank test with *p < 0.05 indicates statistical significance of survival between donor mice from PBS and CC_FLE sE formulated with PCEP-R848, ns indicates not significant for the other groups compared with PBS. Antibody-dependent enhancement (ADE) of DENV-1 (**E**) and DENV-2 (**F**) RVPs using day 42 immune sera. Left: peak enhancement titer (PET); right: magnitude of enhancement at PET. Statistical analysis by Kruskal−Wallis test. **G** In vivo ADE of DENV-2 infection. AG129 mice (*n* = 6/group) were passively transferred with pooled sera from C57BL/6 mice immunized with WT or CC_FLE sE formulated with Adjuplex (as in **E**, **F**). One day later, mice were challenged with 1 × 10$^5$ TCID$_{50}$ of DENV-2 (strain DS210). Survival and terminal lymph node viral loads were determined. Statistical analyses used Mantel−Cox and two-tailed Mann−Whitney U-tests (95% confidence). All assays in (**E–G**) were performed in duplicate. Source data are provided as a Source Data file.

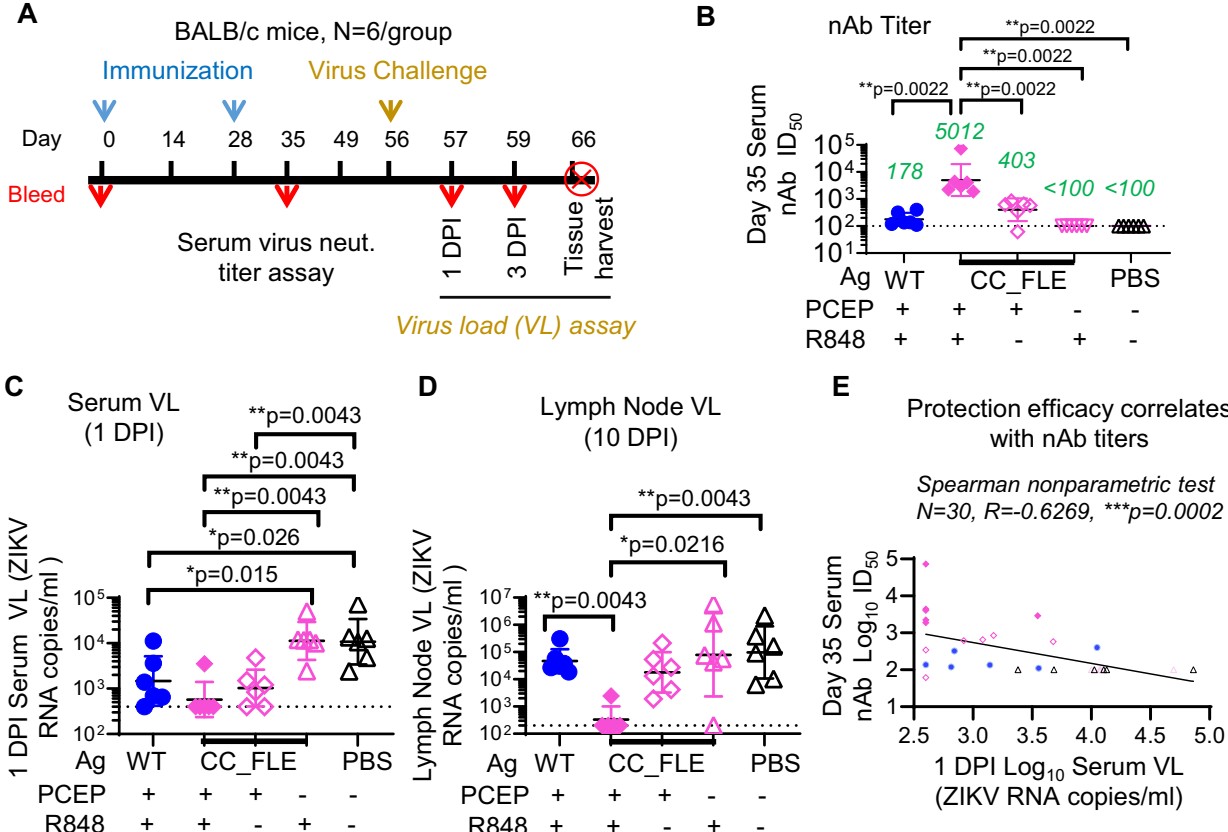

**Fig. 6 | In vivo efficacy study of ZIKV CC_FLE sE formulated in PCEP-R848 adjuvant in BALB/c mice. A** Study design scheme. BALB/c mice (*N* = 6/group) received 20 µg of immunogen formulated in PCEP-R848 were inoculated via subcutaneous (SC) route. ZIKV Puerto Rico 2015 (2 × 10$^6$ TCID$_{50}$) was used for challenge via SC route. **B** Day 35 sera ID$_{50}$ titers (N = 6/group) against ZIKV H/PF/2013 RVP. The geometric mean of ID$_{50}$ for each group is denoted in green font. **C** 1 DPI immune mouse serum virus load (VL) (*N* = 6/group), assayed as ZIKV RNA copy

number/ml. **D** Lymph node VL on 10 DPI (*N* = 6/group). In the assays in (**B**), (**C**), and (**D**), data are presented as geometric mean values +/- geometric SD, with each sample test duplicated. **E** Inverse correlation of serum nAb titers and serum virus load, analyzed using Spearman nonparametric test, ***p* = 0.0002. Unless specified, two-tailed Mann−Whitney U-test, was used for statistical comparison, with 95% confidence. Source data are provided as a Source Data file.

## ZIKV CC_FLE sE immunization confers nearly complete protection in immunocompetent mice

We aimed to evaluate the protective efficacy of our lead immunogen, ZIKV CC_FLE sE, using an immunocompetent mouse model[59,60] that has been validated and widely used in numerous preclinical studies evaluating the efficacy of ZIKV vaccine candidates[61–64]. We immunized immunocompetent BALB/c mice with CC_FLE formulated in the optimal adjuvant system PCEP-R848, using WT sE as a control, followed by ZIKV challenge (Fig. 6A). To evaluate the effect of individual adjuvants, we also immunized mice with CC_FLE formulated in either PCEP or R848. All mice were immunized and challenged as depicted in Fig. 6A.

We chose BALB/c mice as the model animal for vaccine efficacy here based on our in-house observation that after ZIKV challenge, BALB/c mice in general develop higher titers of viremia than C57BL/6 mice (used in our earlier immunogenicity studies), suggesting BALB/c mouse as a suitable model animal for evaluating vaccine efficacy.

We first assessed the nAb titers of the day 35 immune mouse sera. Sera from mice inoculated with WT sE formulated in PCEP-R848 displayed moderate ID$_{50}$ titers (geometric mean = 178) (Fig. 6B). In contrast, sera from CC_FLE sE formulated in PCEP-R848 group showed significantly higher ID$_{50}$ titers (geomean ID$_{50}$ ~ 5,012), 28-fold higher than the WT sE group (**p < 0.01, Mann−Whitney test) (Fig. 6B).

Furthermore, sera from CC_FLE formulated with PCEP-R848 group showed significantly more potent neutralizing activity than sera from animals immunized with CC_FLE formulated with either PCEP (geomean $ID_{50}$ ~ 403, **$p < 0.01$, Mann–Whitney test) or R848 (geomean $ID_{50} < 100$, ***$p < 0.001$, Mann–Whitney test) adjuvant alone (Fig. 6B), indicating the beneficial effect of the supramolecular adjuvant system.

While earlier studies (Figs. 4 and 5) showed similar nAb titers elicited by WT and CC_FLE in C57BL/6 mice, we observed a discrepancy here in BALB/c mice. The variation may stem from the use of different mouse strains. Nonetheless, the nAb titers in BALB/c mice inoculated with monomer WT sE in our study are consistent with historic data from the literature[37]. This suggests that CC_FLE formulated in PCEP-R848 elicits nAb responses superior to WT in the same adjuvant in BALB/c mice, consistent with observations in C57BL/6 mice. Therefore, CC_FLE sE formulated in PCEP-R848 represents an immunization regimen capable of consistently eliciting nAb responses across various mouse strains, outperforming the monomeric WT sE.

To determine if the observed potent nAb titers induced by CC_FLE formulated in PCEP-R848 could confer protection, we challenged the immune mice on day 56 (1 month after the 2nd immunization) (Fig. 6A) with ZIKV Puerto Rico 2015 and monitored serum virus load (VL) thereafter. As a negative control for immunization, sera from mice inoculated with PBS on day 1 post infection (DPI) showed an average of $10^4$ copies of ZIKV RNA/ml. In contrast, 83% (5/6) of mice immunized with CC_FLE sE formulated in PCEP-R848 showed no detectable serum VL (Fig. 6C). WT sE formulated in PCEP-R848 and CC_FLE sE formulated in PCEP alone provided partial protection, with nearly all mouse sera displaying detectable ZIKV RNA (geomean ~ $10^3$ ZIKV RNA copies/ml), which was tenfold lower than PBS-inoculated mice (Fig. 6C). Consistent with the serum VL results at 1 DPI, ZIKV RNA was detectable in lymph nodes (Fig. 6D), in all animal groups (average of $10^4$ ZIKV RNA copies/ml) except the group immunized with CC_FLE sE formulated in PCEP-R848 (Fig. 6D), where only one mouse exhibited detectable ZIKV RNA. Furthermore, there is an inverse correlation between pre-infection serum nAb titers and 1 DPI serum virus load (Fig. 6E), indicating that the nAb response is a key factor contributing to protection efficacy.

## ZIKV CC_FLE sE protection efficacy in pregnancy

We evaluated the efficacy of CC_FLE sE against ZIKV during pregnancy (Fig. S8A). Female BALB/c mice received two immunizations with either WT or CC_FLE sE formulated in PCEP+R848, with PBS serving as the control. Consistent with previous observations, the geometric mean of $ID_{50}$ neutralizing antibody titer in sera from CC_FLE sE−immunized mice was 3240, which was sevenfold higher than that from WT sE−immunized mice ($ID_{50}$ geometric mean = 446) (Fig. S8B). Following immunization, the mice were mated to establish pregnancy and subsequently challenged with ZIKV Puerto Rico 2015 via the retro-orbital route. ZIKV viral loads in serum were measured 1 day post infection (1 DPI). Mice immunized with WT or CC_FLE sE showed 10-fold and 20-fold reductions in viral load, respectively, compared to PBS controls (Fig. S8C, E), indicating that sE immunization reduced ZIKV replication in vivo. On day 8 post challenge, viral loads in maternal uteri were assessed. PBS-inoculated mice showed a geometric mean ZIKV RNA load of $4.7 \times 10^3$ copies/gram tissue (Fig. S8D, E). One mouse from the WT sE−immunized group had detectable uterine ZIKV RNA, whereas none of the mice immunized with CC_FLE sE showed any detectable viral RNA, confirming the protective efficacy of the vaccine in maternal tissues. Furthermore, ZIKV RNA was undetectable in both placentas and fetuses of mice immunized with WT or CC_FLE sE (Fig. S8E). These results demonstrate that CC_FLE sE provided effective protection against ZIKV in both pregnant mice and their fetuses. Taken together, our lead immunogen, ZIKV CC_FLE sE formulated in PCEP-R848 supramolecular adjuvant complex, demonstrates the capacity to elicit potent nAb responses, thereby conferring robust protection in ZIKV-challenged mice and fetus (Figs. 6 and S8), while minimizing the stimulation of ADE-prone Ab responses (Fig. 5). It is likely that CC_FLE sE may elicit a more focused nAb response targeting major nAb epitopes such as the quaternary EDE and DIII LR (Fig. S7), while avoiding the stimulation of non-neutralizing ADE-prone antibody responses. This outcome aligns with the initial rationale behind the structure-based design (Fig. 1).

## High-resolution analysis of ZIKV CC_FLE sE-elicited antibody/B cell response

To delineate the nAb response induced by CC_FLE sE at a high resolution, we utilized a transgenic mouse model known as OmniMouse® (OmniAb, Inc.). These mice lack endogenous $J_H$ and $C_K$ genes, preventing the production of endogenous immunoglobulins. Instead, they carry transgenes containing un-rearranged human antibody Ig light chain kappa and lambda loci, along with recombinant transgenes comprising human $V_H$, $D_H$ and $J_H$ genes with rat heavy chain constant regions. With 44 functional $V_H$, 20 functional $V_K$ and 15 functional $V_L$ genes, OmniMouse animals closely mimic the naïve human B repertoire, facilitating the characterization of human B cell repertoire in response to immunization in preclinical setting.

We immunized two OmniMouse mice with ZIKV CC_FLE sE two and three times, respectively (Fig. 7A). After the second and third immunizations, the mouse sera showed potent neutralization $ID_{50}$ titers against ZIKV H/PF/2013 reporter virus particles, ~600 and 2000, respectively (Fig. S9A, B). The neutralization specificity mapping of OmniMouse immune sera closely resembled that of the dimer-context-specific mAb EDE1-C8 (Fig. 7B and S9C), as well as the sera from CC_FLE sE-immunized C57BL/6 mice (Fig. S7). We subsequently sacrificed the immune mice and employed a FACS-based antigen-specific single B cell sorting and cloning methodology[65–67] developed in-house, to isolate CC_FLE binding mAbs from mouse splenocytes.

In an initial attempt, we utilized ZIKV CC_FLE-PE conjugate as the antigen probe to sort CC_FLE memory B cells binding to CC_FLE at single-cell density (Figs. 7C and S10). We then amplified the Ig heavy (HC) and light chain (LC) encoding genes of each cell. From the obtained 29 cells with both HC and LC amplified, we randomly picked four individual cells with matched HC/LC pairs to express full-length mAbs in human IgG1 form[68] for functional analysis.

As anticipated, all four mAbs, named OZ-D4, OZ-B11, OZ-A1, and OZ-B9 (Table S3), showed strong binding to the ZIKV dimeric sE antigen CC_FLE (Fig. 7D). However, OZ-D4 and OZ-B11 exhibited poor binding to monomeric WT sE (Fig. 7D), resembling the quaternary EDE mAb, EDE1-C8, indicating dimer-context specificity. Additionally, while OZ-A1 showed strong binding to WT sE, it displayed no binding to DIII. Conversely, OZ-B9 bound to both WT and DIII sE, akin to the binding profile of DIII LR mAb Z004 (Fig. 7D). Of note, none of these mAbs displayed binding to E proteins of DENV-1 or -2 (not shown), indicating no cross-reactivity.

To delineate the epitope specificity of the generated mAbs, we performed a BLI-based Ab competition assay, evaluating the competition between the four identified mAbs and a panel of ZIKV reference mAbs (Fig. 7E). The results revealed that these mAbs can be categorized into two competition clusters: (i) OZ-D4 and OZ-B11, which compete mutually and with EDE mAbs, and (ii) OZ-A1 and OZ-B9, which exhibit mutual competition and compete with the DIII CC-loop mAb, ZV64. Notably, OZ-B9 also competes with DIII LR mAb Z004. Based on these results, we speculate that OZ-D4 and OZ-B11 likely target the quaternary epitope EDE, while the epitope of OZ-B9 overlaps with both the DIII LR and DIII CC-loop. The epitope of OZ-A1 may partially overlap with DIII and potentially involve moieties from other domains such as DI.

Importantly, all ZIKV CC_FLE sE-elicited mAbs demonstrated neutralization activity against the ZIKV reporter virus particle (Fig. 7F). OZ-D4 showed an $IC_{50}$ titer of 0.048 µg/ml, comparable to that of

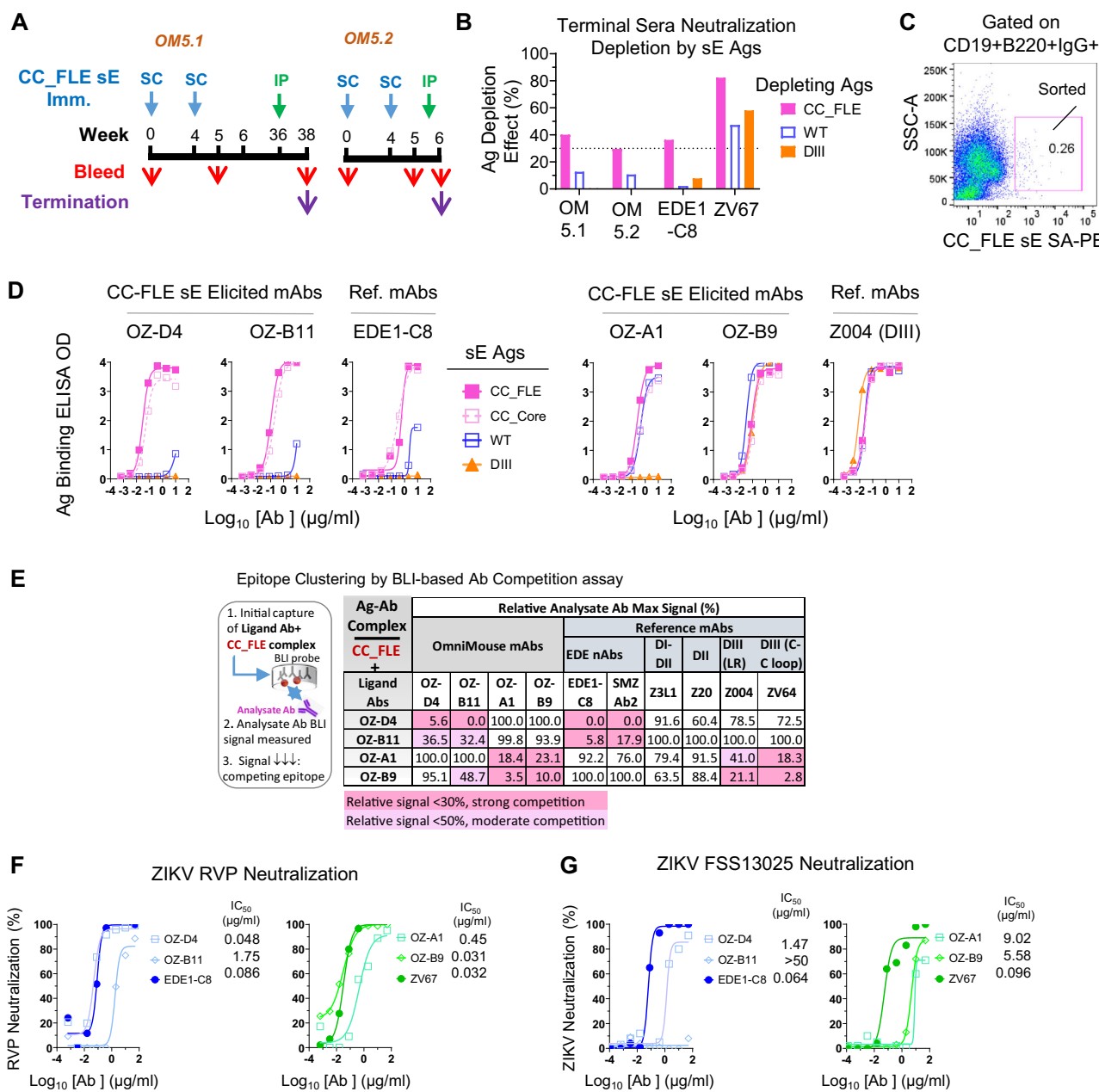

**Fig. 7 | ZIKV CC_FLE sE stimulates nAb responses in OmniMouse animals expressing naïve human Ig loci, with a substantial portion focused on ZIKV E dimer-specific epitopes. A** Immunization schedule. Two female OmniMouse mice, OM5.1 and OM5.2, aged 20–21 weeks, were immunized with CC_FLE sE via subcutaneous (SC) route or intraperitoneal (IP) route at indicated time points. **B** Immune sera neutralization specificity mapping by neutralization depletion with various sE-based antigens (sE Ags). Terminal sera were pre-mixed with sE variant antigens to deplete cognate serum neutralizing antibodies before ZIKV H/PF/2013 RVP incubation, with mAbs, ZV67 (DIII specific) and EDE1-C8 (dimer context specific) as controls. Mock refers to medium containing no antigen added to immune sera or mAb for comparison. **C** CC_FLE sE reactive memory B cell sorting using splenocytes from animal OM5.1. SA-PE, Streptavidin-Phycoerythrin conjugate. **D** ELISA binding profiles of selected CC_FLE-elicited mAbs. Reference mAbs include EDE1-C8 (EDE-specific) and Z004 (DIII-specific). **E** Epitope clustering by BLI-based Ab competition assay. Left: assay scheme; Right: relative CC_FLE sE binding signal after co-incubation of test mAb and competitor mAbs. **F** ZIKV H/PF/2013 RVP, and **G** Authentic ZIKV FSS13025 neutralization mediated by mAbs, assessed by PRNT. In the assays in (**B**, **D**, **F**, **G**), each sample test was duplicated. In (**B**, **D**, **F**), each assay was repeated at least two times. Source data are provided as a Source Data file.

EDE1-C8, while OZ-B9 displayed a potency ($IC_{50}$ = 0.031 µg/ml) similar to that of the DIII LR mAb, ZV67 (Fig. 7F). Consistently, OZ-D4, OZ-A1, and OZ-B9 exhibited neutralizing activity against authentic ZIKV strain FSS13025 in a plaque reduction neutralization test (PRNT) (Fig. 7G), albeit with reduced potency compared to the ZIKV RVP assay, while OZ-B11 showed no detectable neutralizing activity. These results underscore the ability of CC_FLE sE to elicit neutralizing antibody responses targeting the key nAb epitopes, including the quaternary EDE and DIII LR epitopes in OmniMouse mice that express naive

human Ig loci (Table S3). OZ-D4, identified as an immunization-elicited EDE potent nAb, was selected for further studies.

## Structure determination of ZIKV nAb isolated from OmniMouse animal
To gain detailed information on the cognate epitopes of selected nAbs (Fig. 7) isolated from OmniMouse, we attempted to solve the structures of the nAbs complexed with ZIKV sE dimer antigens using cryo-EM single particle reconstruction. Initially, our efforts using ZIKV

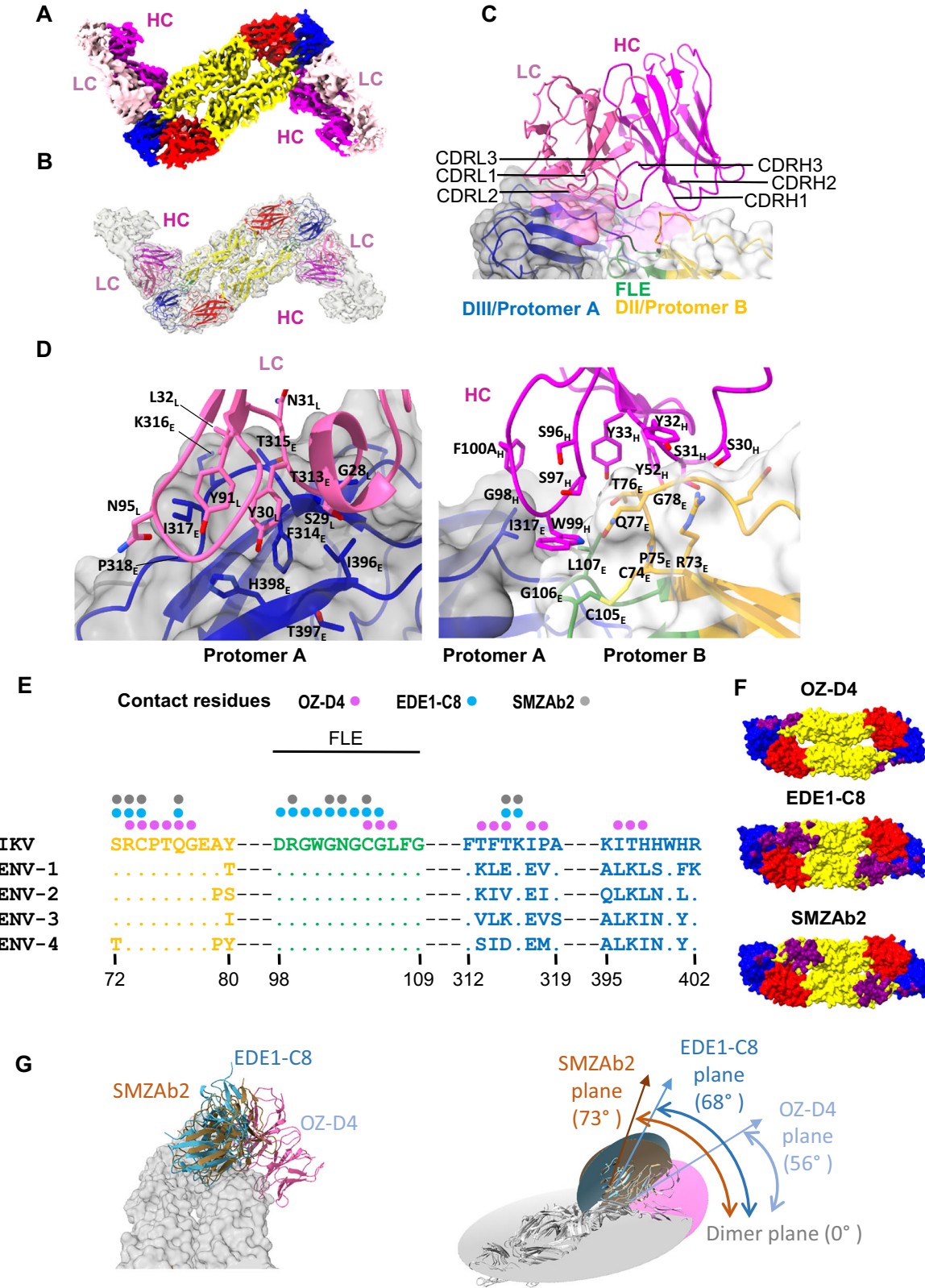

CC_FLE sE as the antigen yielded data of suboptimal resolution, likely due to cryo-EM technical issues. We then employed ZIKV CC_Core sE, a dimeric sE variant bearing the A264C mutation, which presents nAb epitopes effectively (Fig. 2) and exhibits similar binding affinity to OmniMouse mAb as CC_FLE sE (Fig. 7D), as the antigen instead. A structure with a resolution of 3.45 Å was obtained for the OZ-D4: ZIKV CC_Core sE complex (Figs. 8A, B and S11, and Table S4), revealing a

ZIKV E dimer comprised of two protomers (designated as protomer A and B, respectively) in a single conformation bound to two OZ-D4 Fab fragments. Each OZ-D4 Fab binds an E dimer quaternary epitope site encompassing the DII and DIII (Fig. 8B), partially overlapping with the footprint of the reference mAb EDE1-C8. Overall, OZ-D4 binds to DIII of protomer A and DII of protomer B, including part of FLE (Fig. 8C). The OZ-D4 epitope residues on the E dimer include: (i) light chain contact

**Fig. 8 | Cryo-EM structure of OZ-D4 mAb in complex with ZIKV E stabilized dimer CC_Core. A** 3.45 Å resolution cryo-EM map colored by chains of bound OZ-D4 Fabs: heavy chain (magenta), light chain (pink). Density corresponding ZIKV E dimer is shown in grey. **B** Overall structure of OZ-D4:CC_Core complex fitted into the semitransparent Cryo-EM density shown in grey. ZIKV E dimer is colored by domains: domain I (red), domain II (yellow), domain III (blue), fusion loop (green). OZ-D4 colors as in (**A**). **C** Overall view of OZ-D4 binding interface with both pro-tomers (designated as protomer A & B, respectively) of ZIKV E dimer. Semi-transparent surface representation of OZ-D4 epitope on ZIKV E dimer interface: protomer A binding surface is shown in pink, protomer B – in magenta. **D** Cartoon representation of interacting residues of the OZ-D4 light chain (left) and OZ-D4 heavy chain (right) with ZIKV E dimer. **E** ZIKV sE contact residues of OZ-D4 (denoted in magenta) overlapping with prototypical EDE mAbs: EDE1-C8 (PDB: 5LBS) & SMZAb2, denoted in cyan and grey, respectively, in alignment with DENV1-4 sEs. **F** Binding epitopes of OZ-D4, EDE1-C8, and SMZAb2 on the surface of the ZIKV E dimer are shown in purple. **G** Superposition of EDE mAb–sE complexes showing comparison of mAb dimer approach angles. OZ-D4 (pink) in complex with CC_Core, EDE1-C8 (marine) in complex with WT dimer (PDB: 5LBS), and SMZAb2 (brown) in complex with CC_FLE.

residue clusters 313-318 (F314, T315, & I317) and 396-398 (I396, H398) of DIII on protomer A (Fig. 8D, left), (ii) heavy chain contact residues including the end of β-strand b and adjacent loop residues 73–78, the fusion loop residues G106 and L107 (with 80 Å$^2$ buried surface area) on DII of protomer B, and I317 on DIII of protomer A (Fig. 8D, right). Note that hydrophobic residue I317 is engaged in both OZ-D4 heavy/light chain interactions, accounting for nearly 18% of the E dimer/OZ-D4 interface (Fig. 8D), while the other prominent hydrophobic contact residue, F314, contributes to 20% of this interface by engaging the OZ-D4 light chain (Fig. 8D).

OZ-D4 CDR-L1, CDR-L3, and CDR-H3 make the largest contribution to the contact surface area with the ZIKV E dimer. Light chain residues Y30$_L$ and Y91$_L$ interact with F314 and I317, respectively (Fig. 8D, left), while OZ-D4 CDR-H3 residue W99$_H$ is involved in hydrophobic interaction with E dimer residues I317 and L107 at the E protomer interface (Fig. 8D, right). In summary, these data indicate that OZ-D4 has the major features of dimer context dependent EDE-class nAbs (Fig. 8E, and Table S6).

However, the difference in epitope location between OZ-D4 and classical EDE mAbs, including EDE1-C8 and SMZAb2, is well appreciated (Fig. 8E, F, and Table S6). Specifically, OZ-D4 contacting residues 396-398 of ZIKV E are divergent from the corresponding residues on the DENV E protein and are absent in the epitopes of EDE1-C8 and SMZAb2 (Fig. 8E), consistent with the observation that OZ-D4 is not cross-reactive with DENV-1 or DENV-2 E protein. Additionally, the E dimer approach angle utilized by OZ-D4 is distinct from that of classical EDE mAbs. Superposition of mAb–E dimer complexes (Fig. 8G) shows that the angles between the classical EDE mAb planes and the E dimer plane are 68° for EDE1-C8 and 73° for SMZAb2—both close to vertical. In contrast, the angle for OZ-D4 is 56°, which is more lateral. This more lateral approach, to some extent, causes steric hindrance from the viral membrane, consistent with the observation that OZ-D4 exhibits reduced neutralization potency against authentic ZIKV in the PRNT assay (Fig. 7G). Future work using electron microscopy–based polyclonal epitope mapping (EMPEM)[69,70] and isolating additional mAbs from immunized mice will be informative for further characterizing the epitopes and approach angles of EDE-targeting antibodies elicited by CC_FLE sE.

Note that classical EDE mAbs were isolated from DENV- infected individual[71]. Therefore, to our knowledge, OZ-D4 represents immunization-elicited potent nAbs encoded by human Ig gene segments that targets the quaternary neutralizing epitope of ZIKV E protein investigated at atomic resolution.

## OZ-D4 clonal affinity maturation and elicitation by immunization

The elicitation of a specific antibody involves critical steps, including the activation of naïve B cell germline precursor by an immunogen, followed by clonal affinity maturation. This process is characterized by the accumulation of somatic hypermutations (SHMs) in the antibody variable domain genes, driven by affinity selection. These SHM mutations cause divergence from the naïve germline precursor V(D)J sequences, leading to increased affinity for the immunogen[72,73]. Comparison of the heavy chain (HC) and light chain (LC) sequences of OZ-

D4 and the corresponding inferred human germline gene segments, IGHV4-59, and IGLV2-23 (Table S3, and Fig. 9A), respectively, reveals low to moderate level of SHM in the OZ-D4 variable domain genes. The amino acid sequence of the OZ-D4 HC is identical to the germline counterpart, while the OZ-D4 LC shows 8.1% SHM (Table S3, and Fig. 9A). Moreover, most of the residues within the OZ-D4 paratope contacting the sE dimer are identical to the germline version (Fig. 9A), suggesting that extensive accumulation of SHM (e.g., above 10%) is not required for OZ-D4 functionality.

We generated germline-reverted OZ-D4 (OZ-D4_GL) based on the inferred germline sequences to examine the affinity for selected sE immunogens and the contribution of SHM to OZ-D4 function. As expected, the monomeric WT sE shows no binding to either OZ-D4 or OZ-D4_GL, whereas both ZIKV dimeric CC_Core and CC_FLE sEs display high apparent affinity, $K_D$, for OZ-D4 or OZ-D4_GL at pico-molar and nano-molar scale (Fig. 9B), respectively, consistent with the immunogen design rationale that dimeric sE immunogens are superior for engaging and activating EDE epitope-directed naïve B cell precursors over the monomeric sE WT. Additionally, for OZ-D4_GL, CC_FLE sE shows slightly higher affinity, with ~1.5-fold and 3-fold decreases in the dissociation constant ($K_D$) and off-rate ($k_{off}$), respectively, compared with CC_Core sE (Fig. 9B). This enhanced affinity may be attributed to the engineered disulfide bond in CC_FLE sE, which tethers the FLE to the sE N-terminus, improving the presentation of the antibody epitope. Furthermore, OZ-D4_GL can neutralize ZIKV H/PF/2013 RVP, albeit with lower potency than the mature OZ-D4 (Fig. 9C). This observation confirms that the recognition surface between the naïve precursor OZ-D4_GL and the ZIKV E protein is largely preformed prior to immunization. Collectively, our data suggest that dimeric immunogens, such as CC_FLE sE can readily activate the EDE epitope-directed naïve precursor B cells represented by OZ-D4 and drive sufficient affinity maturation to confer protection through immunization.

## Discussion

Here, we employ a structure-based vaccine design approach to create an inter-chain disulfide bond linking the N-terminus with the immunodominant FLE element of flavivirus envelope glycoproteins via G5C/G102C mutations leading to reduced FLE antigenicity in several flavivirus sEs, including JEV, WNV, and ZIKV. Our lead vaccine candidate, ZIKV CC_FLE sE, when formulated in an optimized adjuvant, showed nearly complete protection in immune mice challenged with ZIKV and abolished ADE potential as assessed by in vitro assays and in vivo study. In WT sE, intra-chain disulfide bonds, C3-C30 and C105-C74, connect the N-terminus to Domain I and the FLE to Domain II, respectively[16]. These linkages are crucial for maintaining both local stability and the overall conformation of sE. In close proximity—only 2 to 3 amino acid residues away from one side of these intra-chain disulfide bonds—the G5C/G102C mutations in CC_FLE sE facilitate the formation of a third bivalent linkage by creating an inter-chain disulfide bond between the N-terminus and the FLE (Fig. 1B). This engineered linkage further stabilizes the sE dimer shown by substantially increased Tm by 15 °C (Fig. 2B) and occludes the FLE, as well as the sE N-terminus, which is adjacent to another ADE epitope consisting residues N8-E13 of DENV sE —identical to those in the corresponding regions of

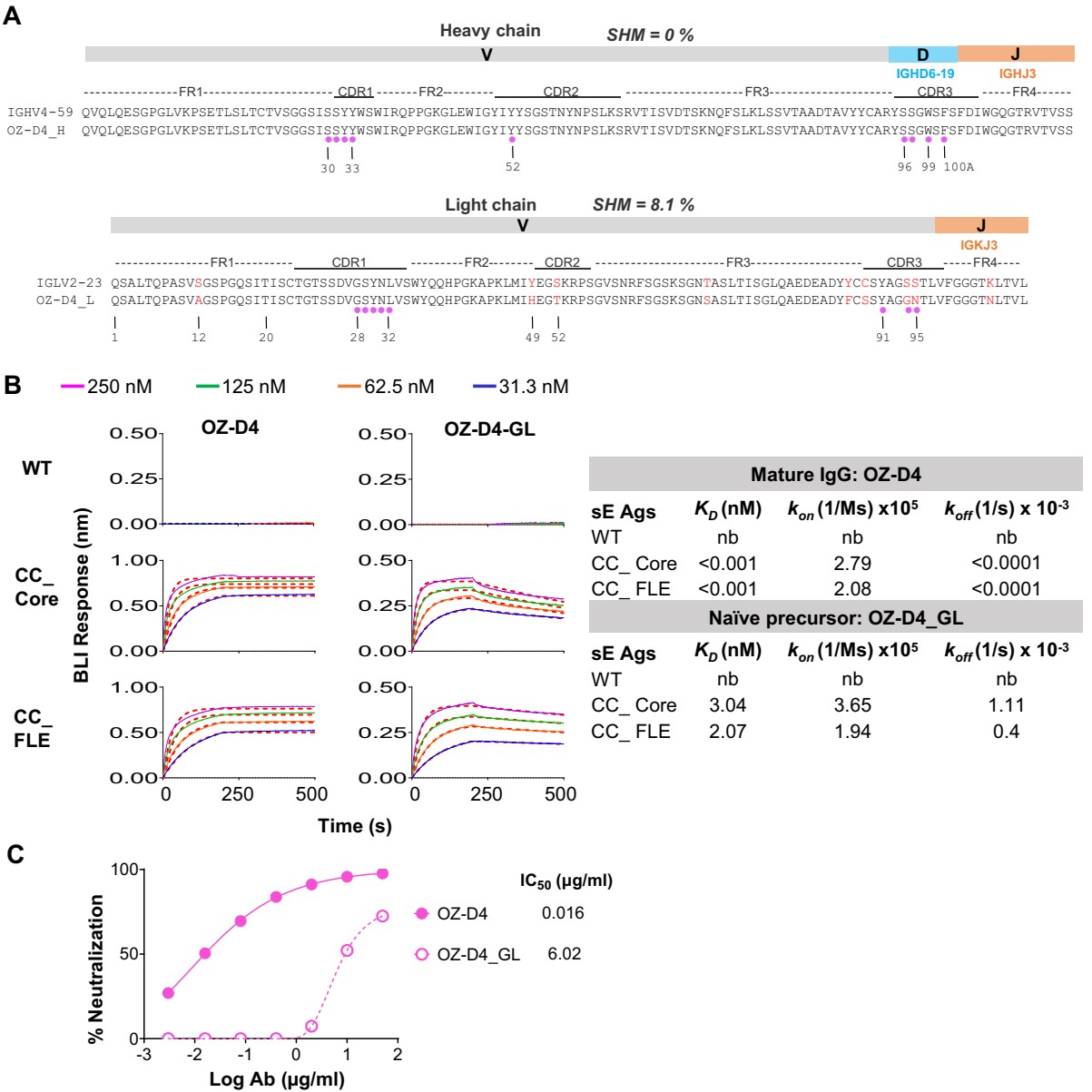

**Fig. 9 | OZ-D4 HC and LC gene amino acid sequence and clonal affinity maturation. A** Alignment of mature OZ-D4 HC (OZ-D4_H) and LC (OZ-D4_L) and the corresponding inferred germline (GL) human Ig V(D)J gene segment sequences. SHM, somatic hypermutation level by amino acid sequence. Divergent residues between the mature OZ-D4 and OZ-D4_GL are in red font. Residues involved in contacting ZIKV sE dimer are denoted with magenta circles. CDRs are delineated using the Kabat system. **B** Kinetics of mature IgG OZ-D4 and the inferred germline version (naïve precursor), OZ-D4_GL binding to ZIKV sE antigens (Ags) analyzed by Bio-Layer Interferometry (BLI). *Left:* BLI binding curves of OZ-D4 and OZ-D4-GL IgG1 molecules; *Right:* dissociation constant ($K_D$), on-rate ($k_{on}$), and off-rate ($k_{off}$) values for each ZIKV sE antigen. nb, no binding. Antibodies were initially captured on anti-human Fc BLI probe, followed by immersing into wells containing sE ranging from 31.3 to 250 nM in concentration. **C** Neutralization activity of OZ-D4 and OZ-D4-GL against ZIKV H/PF/2013 RVP. Each sample test was duplicated, and each assay was repeated at least two times. Source data are provided as a Source Data file.

ZIKV sE[74]. Future studies aimed at delineating the effects of the G5C/G102C mutations on various ADE epitopes would be informative. Our CC_FLE sE design complements previous efforts to reduce the induction of ADE-prone antibodies targeting the FLE on the viral E protein, while expanding its applicability to other flaviviruses such as JEV and WNV. These earlier strategies, distinct from ours, included introducing point mutations in the ZIKV FLE[36] to directly alter its antigenicity, or engineering dimeric sE with reduced FLE exposure[37,41,75]. Such dimeric sE designs involved mutations that facilitate disulfide bond linkages between the FLE and Domain III[37,41], or mutation sets that enhance sE dimer formation[75] to minimize FLE immunogenicity. None of them attempted to tether the FLE to the sE N-terminus.

Since flaviviruses are antigenically related, infection or vaccination with one flavivirus could elicit cross-reactive antibody responses that can potentially not only enhance heterologous viral infection via the ADE mechanism[18,76] but also inhibit the immunogenicity of heterologous vaccines[32–35] administered subsequently through immune imprinting, as demonstrated by recent studies. FLE has been identified as the dominant cross-reactive antibody response epitope responsible for such adverse effects[18,32,76]. Therefore, we anticipate that our CC_FLE vaccine design will also mitigate the immune imprinting effect on other flavivirus vaccines by decreasing the elicitation of FLE-directed, pan-flavivirus cross-reactive antibody responses, which merits further investigation.

Moreover, we have used OminiMouse animals carrying diversified and unrearranged human antibody loci to study sE-elicited B cell responses, enabling the characterization of the human B cell repertoire in response to CC_FLE sE immunization in a preclinical setting. As expected, immune sera neutralization specificity mapping (Fig. 7B and S9) and antigen-specific single B cell sorting and monoclonal antibody cloning analysis (Fig. 7) revealed that dimeric CC_FLE sE is able to elicit potent nAb responses targeting multiple prominent epitopes, such as the quaternary EDE and DIII LR epitopes. This contrasts with the nAb responses elicited by monomeric WT sE, which are limited to DIII epitopes (Fig. S7 and S9). Furthermore, from the CC_FLE sE-immunized human-Ig-loci expressing OmniMouse animal, we isolated OZ-D4, one of the nAbs that displays quaternary EDE epitope specificity confirmed by cryo-EM structural analysis. In addition to certain E protein contact sites similar to the classical EDE nAbs, the OZ-D4: E interface shows unique E contacts, including residues L107 and I317 (Fig. 8E). The high-affinity recognition of dimeric CC_FLE sE by the germline version of OZ-D4, and the low to moderate level of SHM displayed by the OZ-D4 heavy/light chain genes strongly indicate that ZIKV CC_FLE sE may efficiently elicit nAb responses in naïve human B cell repertoires (Fig. 9). This further supports CC_FLE sE as a safe and effective ZIKV vaccine candidate for future advancement. The high-resolution definition of the epitope of the subclass of EDE nAbs represented by OZ-D4 could be exploited to aid future immunogen design for optimizing immunogenicity.

For a subset of the vulnerable population, such as pregnant women and fetuses, sterilizing immunity is desirable to prevent the devastating consequences of ZIKV infection, including malformations and microcephaly in neonates, as well as developmental challenges in children[77]. In this regard, our lead ZIKV vaccine candidate CC_FLE sE could serve as a new starting point toward this goal, given its nearly complete protection efficacy (over 80% of challenged mice showed no detectable virus load in sera and tissues). Recently, a study using recombinant chimpanzee type 7 adenovirus vector to express ZIKV prM/E gene with five mutations in the FLE derived from arthropod-specific flaviviruses showed promising results[40]. In this study, a couple of vaccine candidates demonstrated excellent protection efficacy against ZIKV challenge in immune mice and drastically reduced the elicitation of ADE-prone antibody responses. We reason that combining the selected beneficial FLE mutations from this study with the G5C/G102C mutations in our vaccine candidate, CC_FLE sE, could lead to an even better version of ZIKV E-based vaccine. This new vaccine would not include the prM protein, which has been found to elicit a high degree of cross-reactive antibodies related to ADE[25].

We have observed the synergic effect of co-assembling water-soluble synthetic PPZ-based adjuvant PCEP with the TLR7/8 agonist R848, potentiating the protection efficacy of CC_FLE sE (Fig. 6). This combination increased nAb titers in immune mice over tenfold (Fig. 6B) and reduced tissue virus load for 0.5–2 logs compared to singularly administered PCEP or R848 (Fig. 6D). PPZ adjuvants are known to accelerate onset of the immune response, prolong immunity, and augment and modulate the quality of the immune response[51]. When combined with vaccine antigens, water-soluble PPZ macromolecules can assemble them into supramolecular complexes that bear virus-mimicking features, such as nano-scale dimensions and repetitive display of antigens. Our present study demonstrated that an advanced PPZ macromolecule, PCEP, spontaneously assembles the CC_FLE sE immunogen into complexes with a 60 nm diameter while also displaying a potent TLR-7/8 agonist, R848, in its multimeric form on the same construct. This PCEP-mediated CC_FLE sE co-assembly approach resulted in greatly amplified immunopotentiating activity compared to individual adjuvant components and will be further advanced for the development of other vaccines. Our data indicate that neutralizing antibody responses elicited by CC_FLE sE are key contributors to protection efficacy (Figs. 5D and 6E). However,

whether the formulation with PCEP-R848 stimulates cellular T cell responses that may further enhance in vivo protective effects remains to be investigated.

## Methods

### Structural analysis of flavivirus E proteins and design of inter-protomer disulfide bond

The structure visualization of the ZIKV E protein (PDB: 5JHM), JEV E protein (PDB: 3P54), and WNV E protein (PDB: 3IYW) was performed using Chimera, ChimeraX[78] or Pymol (PyMOL Molecular Graphics System, Version 1.8 Schrödinger, LLC). Buried surface areas (BSAs), accessible surface areas (ASAs), and residue contacts were calculated using the PDBePISA server[79], Chimera[80], or MAPIYA[81]. Inter-protomer disulfide bond design was conducted using Disulfide by Design 2.0[82].

### DNA constructs and protein expression

The DNA encoding ZIKV WT sE, contains amino acid residues 1-404 of the ZIKV E protein from the Brazilian Zika SPH2015 strain (GenBank: KU321639), a prototypical circulating strain during the 2015–2016 Americas outbreak. The JEV sE construct contains residues 1–406 of the JEV E from the SA-14-14-2 strain[83], while the WNV sE construct contains residues 1–400 of WNV E protein[84]. All DNA constructs were synthesized by GenScript (Piscataway NJ) or Twist Bioscience (South San Francisco, CA) and cloned into pMT/BiP Vector Version B (ThermoFisher), with a myc tag at the N-terminus and a 6 × His Tag at the C-terminus. Mutagenesis to construct ZIKV sE dimer variants were performed using standard Gibson assembly cloning with primers containing the mutant sequence for CC_Core (A264C) and CC_FLE (G5C/G102C). For the construction of DI-DIII sE, domain II (residues 51–133 and 195-287) was replaced by two linkers, GGGGS or GGGGSGGGGS, respectively. The DIII sE construct contains residues 303–404. To generate the DIII-NP sE construct, DIII was connected with a ferritin nanoparticle (NP) moiety via a linker. The protein constructs were expressed in *Drosophila* Schneider S2 cells (ThermoFisher, CAT# R69007) according to the manufacturer's instruction. Protein purification was performed as previously described[85], with additional purification and analysis by size-exclusion chromatography using a Superose 6 column or Blue-Native gel analysis.

Plasmids encoding the heavy and light chains of antibodies (Table S5) EDE1-C8, SMZAb2[46], 2A10G6, E16, E53, Z004, Z006, and ZV67, in a human IgG1 expression vector[66,86], and 4G2 and 9E10 in a mouse IgG2 expression vector, were transiently transfected into FreeStyle™ 293F cells (ThermoFisher, CAT#R79007) and purified as described previously[66,87]. E60 was kindly provided by Dr. Michael Diamond.

### Differential scanning calorimetry (DSC)

The thermal transition temperature (Tm) of the ZIKV WT sE and variants were determined by DSC using a MicroCal VP-Capillary DSC instrument (Malvern Panalytics). The protein samples were dialyzed in PBS, pH 7.4, and 400 μl of the sample at concentration of 0.27 mg/mL was loaded into the instrument. PBS buffer was used as the reference solution. The DSC experiments were performed at a scanning rate of 1 K/min under 3.0 atmospheres of pressure. The data were analyzed after buffer correction, normalization, and baseline subtraction using MicroCal VP-Capillary DSC analysis software provided by the manufacturer (Origin 7 SR4 v 7.0522). The assays were repeated at least two times.

### BioLayer light interferometry

Biolayer light interferometry (BLI) was performed using an Octet RED96 instrument (ForteBio, Pall Life Sciences) as described previously[66,87]. The antibody was captured onto anti-human Fc biosensors at a concentration of 10 μg/ml as ligand, with test sample of

sE proteins diluted in twofold series ranging from 250 nM to 62.5 nM unless otherwise specified. Biosensors, pre-hydrated in binding buffer (phosphate buffered saline (PBS), 0.01% BSA, and 0.2% Tween-20) for 10 min, were first immersed in binding buffer for 60 s to establish a baseline. They were then submerged in a solution containing the antibody ligand for 60 s to capture it, followed by a 60-s wash in binding buffer. The biosensors were subsequently immersed in solutions with various concentrations of analyte for 120 s to monitor analyte/ligand association, followed by 120 s in binding buffer to assess dissociation. Binding affinity constants, including the dissociation constant ($K_D$), on-rate ($k_{on}$), off-rate ($k_{off}$), were determined using Octet Analysis software.

### ELISA binding assays

mAbs were coated onto 96-well Maxisorb ELISA plates at 200 ng/well in PBS and incubated overnight at 4 °C. The following day, the plates were washed four times with 300 µL/well of 1× PBST (PBS/0.05% Tween-20). The wells were blocked with a blocking buffer (2% dry milk / 5% fetal bovine serum in PBS) for 1 h at 37 °C. After blocking, the plates were washed again, and antigen proteins were added, starting at 10 µg/ml and diluted in a fivefold series, followed by incubation for 1 h at 37 °C. After another wash step, a 1:5000 dilution of Goat anti-His-HRP conjugate (Jackson ImmunoResearch) in PBST was added to each well and incubated for 1 h at room temperature. The antibody-antigen binding signal was developed by adding 100 µl/well of 3,3′,5,5′-Tetramethylbenzidine (TMB) substrate (Life Technologies) and incubating at room temperature for 5 min, before stopping the reaction with 50 µl of 3% $H_2SO_4$. The optical density (OD) signal was measured at 450 nm to quantify the binding interaction.

### Viruses

The stock of ZIKV FSS13025 (Genbank: JN860885)[88] which is identical or highly related to the America circulating strains was used for challenging AG129 mice that received immune serum transfer. It was the third passage of the virus isolate in Vero cells obtained from the World Reference Center for Emerging Viruses and Arboviruses (WRCEVA) at the University of Texas Medical Branch.

The stock of ZIKV Puerto Rico 2015 strain (GenBank: KU501215.1) that is identical or highly related to the circulating strains during the 2015–2016 Americas outbreak was used for the challenge protection studies of immunized mice. It was prepared as described previously[89]. Briefly, the seed virus obtained from the Division of Vector-borne Diseases, Centers for Disease Control and Prevention, followed by two passages in Vero 76 cells maintained with Leibovitz's L-15 medium supplemented with 10% fetal bovine serum (v/v) 10% tryptose phosphate broth (v/v), L-glutamine at 2 mM, penicillin at 100 units/ml, and streptomycin at 100 µg/ml. For each passage, virus stocks were harvested at 5 days post infection, when ~70% of cells developed cytopathic effect.

DENV strain D2S10 (GenBank accession number JN796245) was originally generated by ten alternating passages of the Taiwanese clinical isolate PL046 between C6/36 Aedes albopictus cells and serum of 129/Sv mice[90]. A passage-three virus was kindly provided by Dr. Sujan Shresta and subsequently passed on a confluent monolayer of C6/36 cells in Leibovitz L-15 medium to produce a tissue-culture stock virus. Tissue culture virus at a tissue culture infectious dose 50% ($TCID_{50}$, $\log_{10}$) titer of ~6.0 was provided for these studies.

### Animal experiments

Animal experiments were carried out in compliance with all relevant U.S. National Institutes of Health regulations and were approved by the Institutional Animal Care and Use Committees (IACUCs) at NOBLE LIFE SCIENCES, INC. (IACUC approval number: SVL-197), BIOQUAL (IACUC approval number 20-026), and University of Maryland School of Medicine (IACUC approval number: #0219005). Unless specified, all animals were housed and handled using Biosafety Level 1 containment. Animal room illumination is automatically controlled in 12-h day/night cycle, with temperature maintained between 20 and 26 °C, and humidity maintained between 30 and 70%. For the immunogenicity study, 8- to 9-week-old female C57BL/6 female mice (Charles River) were inoculated subcutaneously at two sites. Each animal received a single dose of 20 µg protein immunogen in 100 µl of PBS (pH 7.4), containing either 5 µl of Adjuplex (Advanced Bioadjuvants, LLC), formulated according to the manufacture's manual, or polyphosphazene (PPZ) adjuvants[51]. The PPZ adjuvants used were poly[di(carboxylatophenoxy)phosphazene] (PCPP) and poly[di(carboxylatoethylphenoxy)phosphazene] (PCEP), as well as their supramolecular nano-scale complexes with R848 (resiquimod). PPZ-R848 complexes were prepared by mixing either PCEP or PCPP with R848 at 2:1 (w/w) ratio in PBS (pH 7.4), as described previously[57]. Each PPZ adjuvanted formulation contained either 50 µg of PPZ or 70 µg of PPZ-R848 complex (comprising 50 µg PPZ and 20 µg R848). The formation of the ternary complex (CC_FLE-PPZ-R848) was monitored using Asymmetric Flow Field Flow Fractionation (AF4) and dynamic light scattering (DLS). AF4 was performed with a Postnova AF2000 MT series system (Postnova Analytics GmbH, Landsberg, Germany), equipped with two PN1130 isocratic pumps, a PN7520 solvent degasser, a PN5120 injection bracket, and a UV-Visible detector (SPD-20A/20AV, Shimadzu Scientific Instruments, Columbia, MD). A regenerated cellulose membrane with a 10 kDa molecular weight cutoff and a 350 µm spacer was used in a separation micro-channel with both laminar and cross flows of PBS (pH 7.4) as the eluent. Data were processed using AF2000 software (Postnova Analytics GmbH). DLS was conducted using a Malvern Zetasizer Nano series, ZEN3600, and the data were analyzed using Malvern Zetasizer 7.10 software (Malvern Instruments Ltd, Worcestershire, UK). Samples were prepared in PBS (pH 7.4) and filtered through Millex 0.22 µm filters before analysis.

For the serum passive transfer protection efficacy study, 6- to 8-week-old female AG129 mice were infused intraperitoneally (IP) with pooled immune sera 1 h prior to subcutaneous (SC) viral challenge with $1 \times 10^4$ PFU/mouse of ZIKV strain FSS13025. The mice were housed and handled using Biosafety Level 2 containment, and monitored daily for weight loss, health score, and survival for 20 days post challenge.

In the in vivo ADE study, 6- to 8-week-old female AG129 mice were passively transferred with 200 µl of PBS containing 10 µl of pooled donor sera, as described previously[36] via the IP route. One day later, the animals were challenged with $1 \times 10^5$ $TCID_{50}$ of DENV-2 strain DS210, and housed using Biosafety Level 2 containment, monitored daily for weight loss, health score, and survival for 14 days post challenge.

In the vaccine protection efficacy study, 6- to 8-week-old female BALB/c mice (Jackson Laboratory) were inoculated SC over the intrascapular area with immunogen/PPZ adjuvant on study days 0 and 28. On study day 56, 4 weeks post second immunization, the mice were challenged with ZIKV strain Puerto Rico 2015 ($2 \times 10^6$ $TCID_{50}$) via subcutaneous route, a prototype of the circulating strains in Americas, and housed using Biosafety Level 2 containment. Post-challenge, the animals were monitored daily for 10 days for weight loss and clinical manifestation of disease. Blood was collected for serum viral load detection, while spleen, lymph nodes, and brain were harvested on the terminal day for tissue viral load analysis.

In the pregnant mice protection study, immunization of female BALB/c mice was performed in the same manner as described in the vaccine protection efficacy study, with PCEP+R848 as adjuvant. Following immunization, the mice were mated and observed daily for the detection of a copulatory plug (E0.5) indicating pregnancy. At embryonic day 9.5 (E9.5), female mice were challenged with ZIKV Puerto Rico 2015 ($2 \times 10^6$ $TCID_{50}$) via retro-orbital route and euthanized 8 days after infection (E17.5). Maternal sera on 1 and 8 DPI, and tissues including maternal uterus, placentas, and fetuses were collected for ZIKV virus RNA detection by quantitative RT-PCR assay.

In the immunogenicity study with OmniMouse mice, which express naïve human Ig loci, two female OmniMouse mice (OmniAb, CA), designated OM5.1 and OM5.2, aged 20–21 weeks, were immunized with 20 µg of CC_FLE sE formulated in TiterMax Gold (Sigma) adjuvant via the SC route on weeks 0 and 4. Mouse OM5.1 received an additional immunization with 20 µg of CC_FLE sE formulated in Sigma Adjuvant System (Sigma) via the IP route on week 36 and was terminated on week 38. Mouse OM5.2 received another immunization with 20 µg of CC_FLE sE without any adjuvant via the IP route 4 days before termination on week 6.

## ZIKV reporter virus particle (RVP) construction, production, and neutralization assay

To quantify ZIKV mAb neutralization capacity and neutralizing antibody titers in serum samples, we utilized a previously described high-throughput plate-based ZIKV reporter virus particle (RVP) assay[89]. RVPs were generated by complementing a WNV sub-genomic replicon that expresses a luciferase reporter gene (Renilla luciferase) with a plasmid encoding the ZIKV CprME structural proteins (pZIKV/HPF/CprME, ZIKV French Polynesian strain H/PF/2013). Briefly, plasmids pWNVII-Rep-REN-IB and pZIKV/HPF/CprME were co-transfected at a 1:3 ratio into 293 T cells (ATCC, CAT# CRL-3216) using the Fugen6 transfection reagent (Promega). After 48–96 h of incubation at 30 °C, supernatants containing RVPs were collected, filtered through a 0.45-micron filter, and stored at −80 °C. RVPs were titrated on Vero cells (ATCC, CAT#CRL-81) in duplicate to determine the appropriate dilution for the neutralization assay, aiming for ~1–5 × $10^4$ relative luminescence units (RLU) in the absence of serum or antibody, which was 1000 times higher than background (cells only).

To measure neutralization, heat-inactivated serum, starting from a 100-fold dilution with fivefold serial dilutions, was incubated with diluted RVP in Eagle's Minimal Essential Medium (EMEM) containing 10% fetal bovine serum (FBS) for 1 h at 37 °C, before infecting Vero cells at 37 °C and 5% $CO_2$ for 1 h. The next day, cells were lysed with lysis buffer (Promega) for 40 min at room temperature with shaking, followed by the addition of Renilla Luciferase Assay System substrate (Promega) according to the manufacturer's instructions. Luminesce was measured immediately using a Molecular Devices reader. Percent neutralization was calculated based on measurements of wells infected by RVPs in the absence of serum and or uninfected cells. Data were fitted to a 4PL curve in GraphPad Prism 8, with the top and bottom percent neutralization constrained at 100 and 0, respectively. $ID_{50}$ values represented the reciprocal of the serum dilution that resulted in 50% virus entry inhibition compared to RVP alone.

For the serum neutralizing depletion assay, a dilution of pooled sera was pre-mixed with various E variant proteins at a concentration of 0.4 µg/ml for 0.5 h at 37 °C prior to RVP incubation. A dilution of antibodies (ZV67 and EDE1-C8), starting from 0.1 µg/ml, was used as control antibodies. The mixture of RVP with serum or antibody depleted by soluble protein was subsequently added to Vero cells to quantify RVP entry signal (luciferase activity) as described above.

## Plaque reduction neutralization test (PRNT)

The neutralization activity of selected serum samples was measured using a standard plaque reduction neutralization test (PRNT) on VERO cells. Twelve twofold serial dilutions of pooled sera starting from 25-fold dilution, were added to 500 plaque forming units (PFUs) of ZIKV FSS13025 strain in duplicate and incubated for 1 h at 37 °C. The virus/serum mixtures were then added to Vero cells in 24-well plates for an additional hour of incubation, followed by the addition of an overlay of 0.8% methylcellulose. After incubation for 3 days at 37 °C with 5% $CO_2$, the cells were fixed and stained to enumerate plaques using crystal violet staining. $PRNT_{50}$ values were determined as the serum dilution that resulted in 50% of the number of plaques obtained with no serum treatment as control.

## ADE assays

In vitro ADE assays were performed following a published method[91]. Briefly, serial dilutions of heat-inactivated sera, starting at a 60-fold dilution, were mixed with DENV-1 RVPs (Western Pacific strain with GenBank accession number U88535.1) or DENV-2 RVPs (16681 strain with GenBank accession number M84727.1), produced as described above by replacing pZIKV/HPF/CprME with pDENV/CprME. The mixture was incubated for 1 h at 37 °C in duplicate prior to addition to K562 cells (ATCC, CAT#CCL-243) for 2 days. Cells were lysed, and RVP infection was measured for Renilla Luciferase signal. To normalize the magnitude of enhancement across independent experiments, results are displayed relative to the maximum infectivity observed with a control cross-reactive WNV mAb E60[92] run in parallel.

## Quantitative RT-PCR assay for ZIKV and DENV RNA

The quantity of ZIKV or DENV RNA copies per milliliter of plasma or per gram of tissue was determined using the TAQMAN assay[93]. Briefly, viral RNA was extracted from serum using the QIAamp MinElute Virus Spin kit (Qiagen, Frederick, MD) or from tissue samples using the RNASTAT-60 kit (Tel-Test Inc., Friendswood, TX). The RT-PCR assays were performed using the SensiFAST Probe Lo-ROX One-Step Kit (Bioline BIO-78005, Taunton, MA). Primers and probe were designed to amplify a conserved region of the ZIKV capsid gene from the BeH815744 strain, including the primers ZIKV PR15 FWD (GGAAAAAAGAGGCTATGG AAATAATAAAG) and ZIKV PR15 REV (CTCCTTCCTAGCATTGATTAT TCTCA), and the probe ZIKV PR15 PRB (AGTTCAAGAAAGATCT GGCTG), as described previously[93]. The sequences for the DENV primer/probe set, which target a conserved region of the polyprotein gene, are as follows: Denv-2 16681 FWD (GCGGGAAAGACGAAGAGA-TAC), Denv-2 16681 REV (CTCTAAGGGCTTCCTCCATTTC), and Denv-2 16681 Probe (5′-/56-FAM/CTTCCGGCC/ZEN/ATAGTCAGAGAAGCT/ 3IABkFQ/−3′).

To generate a control for the amplification reaction, RNA was isolated from the applicable virus stock using the same procedure. The standard curve was used to calculate the ZIKV or DENV RNA copy number of each test sample, conducted in triplicate.

## Flow cytometry & Ag-specific B cell sorting

Cryopreserved OmniMouse animal splenocytes were quickly thawed in a 37 °C water bath, followed by treatment with DNase I (10,000 µ/ml, Roche, 1000-fold dilution) in RPMI 1640 with 10% FBS at room temperature for 5 min, and washed with chilled PBS. After staining with Aqua Dead Cell Staining dye (Life Technologies, Cat# L349660, LOT#1941440, 400-fold dilution) in PBS for 15 min, followed by two steps of staining procedure, each lasting 1 h, cells were washed with PBS between these two staining procedures. Since OmniMouse mice carry a rat Ig constant region encoding sequences adjacent to the human heavy chain variable gene segments, most of the class-switched B cells would bear rat IgG constant region. The first staining medium contains polyclonal goat anti-rat IgG PE/Dazzle 594 conjugate (Biolegend, CAT#405432, LOT#B319517, 100-fold dilution) to identify class-switched B cells.

The second staining medium contained a cocktail of antibodies and antigen for identifying antigen-specific B cells, as described previously with minor modifications[67]. Briefly, a cocktail of antibodies diluted in RPMI 1640/10% FBS, which contains CD3- PerCP/Cy5.5 (clone 17A2, Biolegend, CAT# 100218, LOT#B233420, 500-fold dilution), F4/80-PerCp/Cy5.5 (clone BM8, Biolegend, CAT#123128, LOT#B222447, 1000-fold dilution), Gr1-PerCp/Cy5.5 (clone RB6-8C5, Biolegend, CAT#108428, LOT#B245570, 2000-fold dilution), CD19-APC-Cy7 (clone 1D3, BD Pharmingen, CAT#557655, LOT#6070644, 500-fold dilution), B220- Alexa Fluor® 700 (clone RA3-6B2, BioLegend, CAT#103232, LOT#B242183, 2000-fold dilution), mouse IgG2a-FITC (clone RMG2a-62, Biolegend, CAT#407106, LOT#B199948, 500-fold dilution), mouse IgG2b-FITC (clone R12-3, BD Pharmingen, CAT#

553395, LOT#7096543, 500-fold dilution), mouse IgG1-FITC (clone A85-1, BD Pharmingen, CAT#553443, LOT#7020839, 500-fold dilution), mouse IgD-Pacific Blue (clone RA3-6B2, BioLegend, CAT#405712, LOT#B209676, 1000-fold dilution), and CC_FLE–streptavidin-PE conjugate (at 2.5 µg/ml) was used to stain the cells. CC_FLE-Avi carrying an Avi-tag at the c-terminus were biotinylated with BirA 500 biotin ligase (Avidity, AviTag™ Technology, Aurora, CO), followed by mixing with streptavidin-PE conjugate (Invitrogen, CAT# S21388, LOT#1784902) at 1:1 molar ratio to form CC_FLE–streptavidin-PE conjugate as described previously[66]. As shown in Fig. S10, the stained cells were applied to a BD FACS Aria III cell sorter (BD Biosciences) to sort for antigen-specific class-switched B cells with BD FACSDiva software V8.02 using the phenotype of CD19+/B220+/CD3−Gr1−F4/80−/mouse IgD−IgG−/rat IgG+/CC_FLE+ into 96-well plates at single-cell density, followed by single-cell reverse transcription and PCR reactions to amplify human Ig V(D)J gene segments as previously described[86,94,95]. The sequences of the PCR primers are provided as Supplementary Data 1 file in the Supplementary Information File folder.

The resultant human Ig gene segments were analyzed by IMGT/V-Quest using the human Ig germline reference database, with the variable domains of selected heavy/light (VH/VL) chain matched clones being cloned into corresponding expression vectors, as described previously[86], followed by co-transfection of VH/VL expression plasmids into 293F cells[66] to reconstitute human antibodies in IgG1 format. The expressed human antibodies were purified by Protein A Sepharose columns (GE Healthcare). To produce an antibody Fab fragment, the heavy chain variable domain was cloned into the Fab expression vector described previously[96], followed by co-transfection with light chain expression vector in 293F cells and purification using cOmplete His-Tag Purification Resin (Roche).

### OmniMouse Zika E-specific mAb competition assay
The following experiment was performed using Bio-Layer Interferometry (BLI) with an Octet RED96 platform (ForteBio; Pall Life Sciences) to determine if antibodies compete for binding to ZIKV E-dimer. ZIKV E-dimer CC-FLE, purified by size exclusion chromatography at 20 nM, was immobilized on the pre-wet Octet® Ni-NTA (NTA) Biosensors (Sartorius) for 3 min, followed by a wash in 1× kinetics buffer (1× PBS, 0.2% Tween-20). The biosensors were then submerged in wells containing the ligand mAb at 10 µg/ml in kinetics buffer for 10 min, followed by a 60-s wash step. Subsequently, the biosensors were immersed in wells containing analysate mAb at 10 µg/ml in kinetics buffer for 10 min, followed by a dissociation step in which the biosensors were held in 1× kinetics buffer for another 10 min. In each step, BLI signals were collected at 24 °C with orbital shaking at 1000 rpm in 96-well black flat bottom plates (Greiner, VWR). The maximal analysate mAb signals were obtained in the absence of any ligand mAb (kinetics buffer only). The relative analysate mAb binding signals were calculated as 100*(Analysate mAb signal in the presence of ligand Ab/ Maximal analysate mAb signal in the absence of ligand Ab).

### Cryo-EM sample preparation, data collection, model refinement and analysis
SMZAb2 Fab was incubated with the ZIKV CC_FLE sE protein at a 2:1 molar ratio per E protomer overnight at 4 °C. The concentrations of the CC_FLE sE: SMZAb2 complex, or unliganded JEV and WNV CC_FLE sEs were adjusted to 1 mg/ml in PBS, and 3 µl of the sE was added to a Quantifoil grid (R2/2 Cu 200 mesh or R1.2/1.3 Cu 300 mesh; Electron Microscopy Services) that had been freshly glow-discharged using a PELCO easiGLOW (Ted Pella). Samples were vitrified in 100% liquid ethane using a Mark IV Vitrobot (Thermo Fisher) at 100% humidity and 12 °C. The micrographs were collected on a ThermoFisher-FEI Glacios transmission electron microscope operating at 200 kV, using custom

scripts in SerialEM software version 4.1[97]. Movies were obtained on a Gatan K3 direct electron detector at a magnification of 56,200×, using a 0.4−2.6 µm or 0.5−2.7 µm defocus range. The micrographs were processed with cryoSPARC version 3.3.0[98]. For the CC_FLE sE: SMZAb2 complex, a total of 141,948 particles were selected for final local 3D refinement with C2 symmetry applied, resulting in a reconstruction at 4.09 Å resolution. For the JEV CC_FLE sE, the final 3D reconstruction and refinement of 172,829 particles with imposed C2 symmetry resulted in a 4.18 Å resolution map. The structure of the JEV E dimer (PDB: 3P54) was used as the initial template. The individual chains of the initial model were docked into Cryo-EM density maps with UCSF Chimera (version 1.18)[80], followed by the iterative cycles of building and refinement with Coot (version 0.9.8.1)[99], Rosetta (version 3.13)[100], and Phenix (version 1.20)[101]. For the WNV CC_FLE sE, the final 3D reconstruction and refinement of 14,474 particles with imposed C2 symmetry resulted in a 7.96 Å resolution map. No attempts to build a structural model of WNV CC_FLE sE were made.

The OZ-D4 Fab was incubated with ZIKV CC_Core sE in a 1.5:1 molar ratio (per E protomer) overnight at 4 °C. The concentration of the complex was adjusted to 1 mg/ml in PBS, and followed by grid preparation, vitrification, and data collection as described above. The final local 3D refinement of 710,140 particles with imposed C1 symmetry resulted in a 3.45 Å global resolution map (Fig. S11).

The structure of ZIKV E dimer (PDB: 5LBV) was used as the initial template for ZIV CC_Core. The OZ-D4 Fab variable region was generated with SAbPred[102]. The individual chains of the initial models were docked into Cryo-EM density maps with UCSF Chimera[80], followed by the iterative cycles of building and refinement with Coot[99], Rosetta[100], and Phenix[101]. The structure building, refinement, and visualizations of the E: mAb complex were performed using similar tools and approaches described earlier.

### Statistical analysis
All statistical analyses are performed using GraphPad Prism V10.1.0, unless otherwise specified. Significance thresholds are set at $p < 0.05$.

Two-tailed Mann−Whitney U tests (95% confidence level) are used for comparisons of serum neutralizing titers (Figs. 5B and 6B), viral loads (Figs. 5G, 6C−D and S8B−D), and $ID_{50}$ titers (Fig. S8B−D). Serum ADE for DENV-1 and DENV-2 (Fig. 5E−F) is analyzed using a one-way ANOVA (Kruskal−Wallis) test. Survival curves (Fig. 5D, G) are compared using the Mantel−Cox log-rank test. Health scores (Fig. S6) are analyzed using multiple $t$ tests, and correlations between serum neutralizing titers and $log_{10}$-transformed viral loads (Fig. 6E) are assessed using the Spearman nonparametric test.

### Reporting summary
Further information on research design is available in the Nature Portfolio Reporting Summary linked to this article.

## Data availability
The structural coordinates data and/or cryo-EM map generated in this study have been deposited in the RCSB Protein Data Bank (PDB) and the Electron Microscopy Data Bank (EMDB) database, respectively, under the following accession numbers: ZIKV CC_FLE sE:SMZAb2 complex, PDB: 9OD2, EMD: 70338; JEV CC_FLE sE, PDB: 9PL9, EMD: 71715; WNV CC_FLE sE, EMD: 71727; and ZIKV CC_Core sE:OZ-D4 Fab complex, PDB: 9PM6, EMD: 71728. The nucleic acid sequences of OmniMouse animal-derived ZIKV mAbs have been deposited in the GenBank/NCBI database, with the following accession numbers: OZ-A1 VH, GenBank: PQ465206; OZ-A1 VK, GenBank: PQ465207; OZ-B9 VH, GenBank: PQ465208; OZ-B9 VK, GenBank: PQ465209; OZ-B11 VH, GenBank: PQ465210; OZ-B11 VK, GenBank: PQ465211; OZ-D4 VH, GenBank: PQ465212; OZ-D4 VL, GenBank: PQ465213. The other data generated in this study are provided in the Supplementary Information/Source Data file. Source data are provided with this paper.

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

## Acknowledgements

We thank Dr. Michael Diamond for providing FLE-specific mAb E60 as a reference reagent. We also thank the Translational Laboratory Shared Service (TLSS) of the University of Maryland Greenebaum Comprehensive Cancer Center for housing and supporting the OmniMouse immunization study. This work is partially supported by funds from the University of Maryland Strategic Partnership (MPower), the Maryland Innovation Initiative Project #0323-010, NIH grants R01AI102766, R01AI136756, P01AI157299 and R01AI175439, the International AIDS Vaccine Initiative, the James B. Pendleton Trust, and the National Cancer Institute-Cancer Center Support Grant (CCSG) - P30CA134274). IBBR CryoEM Center is supported by the National Institute of Standards and Technology (DoC grant 70NANB21H105) and by the University of Maryland Strategic Partnership: MPowering the State. TCP is supported by the Vaccine Research Center, NIAID, NIH. His contributions were made as part of his official duties as NIH federal employees, are in compliance with agency policy requirements and are considered Works of the United States Government. However, the findings and conclusions presented in this paper are those of the authors and do not necessarily reflect the views of the NIH or the U.S. Department of Health and Human Services.

## Author contributions

Conceptualization, Y.W., A.G., A.A., X.S., T.F., E.P., Y.L.; data curation, Y.W., A.C., S.K., A.G., X.S., S.B., Y.L.; formal analysis, Y.W., A.G., X.S., S.B., R.T.W., S.K., A.K.A., E.P., Y.L., T.C.P.; funding acquisition, Y.L., A.K.A., D.J.W.; investigation, Y.W., A.M., C.-I.C., S.J., T.-J.Y., A.C., A.G., X.S., S.B., A.L.C., A.T., J.G., R.S., J.W., H.A., S.K., Y.-J.S., D.L.V., S.H., C.I., A.K.A., E.P., Y.L.; methodology, Y.W., A.M., A.C., A.G., A.C., A.T., S.K., A.A., X.S., S.B., Y.L.; animal studies, A.C., A.T., R.S., J.W., S.K., C.I., R.G.L.; supervision, Y.L., A.K.A., E.P., R.T.W.; writing—original draft, Y.W., Y.L.; writing—review & editing, Y.W., A.G., A.K.A., Y.L., X.S., S.B., S.H., D.J.W., E.P., T.C.P.; All authors have read and agreed to the published version of the manuscript.

## Competing interests

Y.L., Y.W., A.G., and A.K.A. are co-inventers of patent application titled "Engineered Flavivirus Envelope Glycoprotein Immunogenic Compositions and Methods of Use" (PCT/US2023/076441). Y.L. and X.S. are co-inventers of patent application titled "Antibodies for Zika Virus" (PCT/US2023/083586). The remaining authors declare no competing interests.

## Additional information

Yimeng Wang [1,14,15], Andrey Galkin[1,2,15], Xiaoran Shang[1,2,15], Alexander Marin[1], Shaohua Jin[1,2], Ting-Juan Ye [1,2], Shridhar Bale[3], Chi-I Chiang [1], Ananda Chowdhury[1], Agnes L. Chenine[4], Ashley Turonis[4], Jack Greenhouse[5], Rebecca Stone[5], Jaclyn Wear[5], Swagata Kar[5], Hanne Andersen [5], Yan-Jang S. Huang [6], Dana L. Vanlandingham[6], Stephen Higgs[6], Rena G. Lapidus[7], Thomas Fuerst [1,8], David J. Weber [1,9,10,11], Richard T. Wyatt[3], Christel Iffland[12], Theodore C. Pierson[13], Alexander K. Andrianov [1], Edwin Pozharski [1,9,10,11] ✉ & Yuxing Li [1,2,10,11] ✉

[1]Institute for Bioscience and Biotechnology Research, Rockville, MD, USA. [2]Department of Microbiology and Immunology, University of Maryland School of Medicine, Baltimore, MD, USA. [3]Department of Immunology and Microbiology, The Scripps Research Institute, La Jolla, CA, USA. [4]Integrated BioTherapeutics, Inc., Rockville, MD, USA. [5]BIOQUAL, Inc., Rockville, MD, USA. [6]Biosecurity Research Institute and Department of Diagnostic Medicine/Pathobiology, College of Veterinary Medicine, Kansas State University, Manhattan, KS, USA. [7]Department of Medicine, University of Maryland School of Medicine, Baltimore, MD, USA. [8]Department of Cell Biology and Molecular Genetics, University of Maryland, College Park, MD, USA. [9]Department of Biochemistry and Molecular Biology, University of Maryland School of Medicine, Baltimore, MD, USA. [10]Center for Biomolecular Therapeutics, Rockville, MD, USA. [11]Center for Biomolecular Therapeutic, Baltimore, MD, USA. [12]OmniAb, Inc., Emeryville, CA, USA. [13]Vaccine Research Center, National Institute of Allergy and Infectious Diseases, Bethesda, MD, USA. [14]Present address: ReVacc, Inc., Cambridge, MA, USA. [15]These authors contributed equally: Yimeng Wang, Andrey Galkin, Xiaoran Shang. ✉e-mail: epozhars@umd.edu; yuxingli@umd.edu

