## [Transparent Peer Review file · Nature Communications]

Rational Design of Flavivirus E Protein Vaccine Optimizes Immunogenicity and Mitigates Antibody Dependent Enhancement Risk

Corresponding Author: Dr Yuxing Li

Editorial Note: Parts of this peer review file have been redacted as indicated to avoid any copy right infringement.

Version 0:

Reviewer comments:

Reviewer #1

(Remarks to the Author)

Wang and colleagues present in this paper the characterization of an engineered dimer of the Zika virus (ZIKV) envelope protein (E). The Flavivirus E dimer has been shown to be target of potent cross-neutralizing dengue and Zika antibodies (E-dimer epitope antibodies – EDE). The dimer also hide the fusion loop epitope (FLE) which is targeted by antibodies that cause antibody-dependent enhancement (ADE) of infection. FLE antibodies are cross-reactive but poorly neutralizing and ADE is a risk factor for severe pathology during Flavivirus secondary infection. EDE antibodies instead are cross-reactive and highly neutralizing, and several strategies have been developed to engineer antigens able to expose EDE epitopes and hide FLE epitopes. Here, the authors have introduced a G5C/G102C mutations (CC_FLE construct) that generate a disulphide bond between the N-term of domain I (DI) of one protomer and the fusion loop (FL) of the opposite protomer resulting in a locked ZIKV E dimer. They compare this construct to an E-monomer and a E-dimer with a single mutation A264C (CC_Core construct). The latter construct has already been reported in the literature (doi: 10.1074/jbc.RA119.007443; doi: 10.1038/s41598-017-01097-5). The three constructs have been tested for antigenicity and protection in an AG129 mouse and in immunocompetent mice model. In addition, the authors used the CC_FLE antigen to immunize Omni-mouse animals (carrying human Ig-loci) from which 4 antibodies have been cloned and one EDE-like humanized antibody (OZ-D4) has been characterized. The engineered dimer alone or in complex with OZ-D4 has been structurally characterized. The authors gathered an impressive amount of data that however need to be thoroughly discussed in comparison to the already published constructs to highlight the added value of the proposed strategy.

Immunological cross-reactivity between Flaviviruses circulating in the same endemic population makes vaccine design challenging because the antigen to be used in a vaccine formulation, in addition to expose protective epitopes, needs to mask enhancing epitopes and avoid uncovering cryptic epitopes. The strategy to engineer stabilized E-dimer for dengue or Zika has been first proposed by Rouvinski et al. in 2017 (doi: 10.1038/ncomms15411) and used for vaccine design and protection studies by several authors (doi.org/10.1016/j.jbc.2022.102079; doi.org/10.1038/s41590-019-0477-z; doi.org/10.1038/s41590-021-00966-6; doi 10.1074/jbc.RA119.007443).

From the structural point of view there are few points that are not clear:

1) How the introduced mutations affect the overall structure of the E-dimer? In the E dimer, the N-term of DI, leans towards DIII and the DI-150loop is juxtaposed to the FL. In the TBE dimer, for example, amino acids H5 and G102 form an electrostatic bond only at low pH when the 150-loop is in an “open lid” position away from the FL, but at neutral pH the 150-loop is in a “closed lid” position interacting with G102 while the DI N-term shifts slightly away from FL (doi.org/10.1038/s41467-022-31111-y). The newly created C-C bond between G5 and G102 would force DI N-term to stay close to the FL and affect the 150-loop position and probably also DIII overall conformation. From figure S2A and S2C The CC_FLE dimer looks open in the middle with an altered intradimer interface and a bulgy DIII. The PDB model shows a more closed dimer but the fitting of the model in the density map could be slightly adjusted to assure a more precise reconstruction (within the limits of the relative low resolution).

How this structure overlaps with the unlocked dimer (for example from the cryoEM structure of the Zika or JEV virus)? In figure S2D the map density is barely visible, a more intense colour would help to appreciate the amino acids fit in the density. X-ray crystallography for both locked dimers would help in defining the impact of the mutations on the overall structure of the dimer. Structure of the dimer in complex with known human antibodies would allow to verify if the protective

epitopes are correctly exposed.

2) What is the rationale for the strategy reported in lines 169-175? Has the structure of DI-DIII construct been analyzed? DI may not be correctly folded, and this antigen would focus the response only to DIII and probably also to DIII epitopes that usually are not exposed in the dimer.

3) Fig.8: The footprint of OZ-D4 antibody does not overlap with the ones of the EDE antibodies. OZ-D4 binds DII and DIII but not DI (differently from the EDEs) and approaches the dimer from the side. This binding would clash with the herringbone arrangement of the dimer on the virus. A docking of this antibody on the cryoEM of the Zika particle would allow to check the accessibility of the E dimer to OZ-D4. Neutralization has been tested with the RVP system but not with authentic virus. It would be interesting to verify the affinity of OZ-D4 for the dimer in the context of the herringbone organization.

For the antigenicity studies:

1) Could the authors clarify the epitope targeted by antibody SMZAb2? Is this the antibody isolated by Magnani et al. (doi: 10.1126/scitranslmed.aan8184)? SMZAb2 in this publication has been classified as an FL antibody, is binding in ELISA to a ZIKV E protein and is competing with FL antibody 4G2. Here the authors classify SMZAb2 as an EDE that does not recognize the WT monomer construct (Fig. 2C). In addition to EDE-C8 have the authors tested EDE-C10? This antibody has shown to be more sensible to the constraints induced in the E-dimer by engineered C-C bonds and could give information about the flexibility of the C5/C102 E-dimer.

2) Some of the FLE antibodies, like 2AG106, bind G102. Have the authors considered the possibility that the lack of binding of the CC_FLE antigen could be due to the mutation of this amino acid more than a general non-exposure of the FL?

3) Why the binding of FLE antibodies (Fig.2C) is different between WT and CC_Core? The FL should be exposed equally in both constructs because the dimer is "breathing".

4) The neutralization data reported in Fig.4B are not clear. How C264 mutation would affect antigenicity? The impact of this mutation on the structure does not explain the different response obtained compared to WT construct. Also, DI-DIII and DIII construct bind similarly to DIII antibodies, but the neutralization profile is different. Have these sera been tested against authentic virus? May be the organization of the E protein on the RVP plays a role.

5) Fig.5G: How was the amount of serum to be passively transferred chosen? How would a positive control (Virus + FLE antibody at enhancing concentration) behave in this system?

6) Fig.S6A: For the depletion experiments graphs, I would add in the legend a column indicating the Abs specificities left (FLE, DIII, EDE). This would facilitate the reading of the graphs.

7) I would move Fig.S6B to main Fig.6 because it shows the landscape of antibody specificities elicited by the different antigens. I would give more information on how to read the graph. My understanding is that a high Y axis value corresponds to a drop of neutralization activity (i.e. the antigen depletion has removed antibodies important for neutralization). It is interesting that the sera obtained from WT immunization show the highest reduction in neutralization when depleted with the CC_FLE and not when depleted with WT. It would suggest that the immunization with WT has induced also some EDE antibodies (may be because the soluble E protein in solution is in monomer-dimer equilibrium and can also expose dimer epitopes?). These antibodies specificities should be analyzed more in detail and discussed.

8) The OmniMouse model is interesting. Why have the authors not performed a comparison between WT and CC_FLE antigens? I see the interest in cloning specific antibodies from these mice, but it would be also informative to look at the B cell repertoire induced by the two types of antigens to appreciate the superiority of the CC_FLE antigen.

9) Why was the OmniMouse 5.2 terminated at 6 weeks p.i., much earlier than the OmniMouse 5.1 (38 weeks)? From the shape of the neutralization curves, the sera do not show a strong neutralizing activity. Have the sera been tested for presence of FLE antibodies?

10) It is interesting that the complex of OZ-D4 with CC_FLE did not result in high resolution data while the complex with CC_Core did. The antibody binds the two antigens with comparable affinity (Fig.7D).

Minor points:

Line 173: What is (32-34) referring to? The reference number does not match with "DIII immunogenicity" described in the text.

Lines 179-181: It would be informative to show the SEC of profile of the proteins after affinity purification (may be as a supplementary figure) to appreciate the heterogeneity of the forms produced by the constructs.

Lines 191-192: Please list all the antibodies that have been tested with their epitope specificity and recognition of the antigens tested. A figure and a summary table in supplementary data would be informative.

Lines 196-199: This sentence is redundant. These results have been presented in the previous paragraphs.

Fig.5: B and C show neutralization assays results (RVP and authentic virus). I would plot both in the same way (either ID50 or % of neutralization).

Line 364: the ID50 titer increase is not 2 logs (5012 vs 178 = 28 folds).

Lines 399-400: I do not understand the sentence "surpassing the nascent WT sE".

Fig.8: 8A – show the HC and LC chains with more contrasting colors. 8F – show a top view of the dimer to highlight the antibody footprint. The actual side view is not informative.

Reviewer #2

(Remarks to the Author)

The present manuscript by Wang et al presents a novel vaccine design strategy targeting the E protein of the Zika virus (ZIKV), aimed at bolstering immune responses and mitigating the risk of Antibody-dependent enhancement (ADE). The authors have engineered two mutations, G5C/G102C, into the flavivirus E protein, resulting in the formation of inter-chain disulfide bonds and the creation of E-dimers. The E-dimer formation not only augments the immunogenicity of the vaccine

candidate but also masks the ADE-associated site FLE, thereby diminishing the likelihood of ADE. The vaccine candidate, CC-FLE sE, is primarily constructed based on the ZIKV E protein sequence. Upon immunization in mice, it elicits potent neutralizing antibodies that confer protection against ZIKV infection and attenuate the risk of ADE following infections with various flaviviruses. Furthermore, the article delineates the structural and functional analysis of the E-dimer-dependent epitope (EDE) monoclonal antibody OZ-D4, highlighting the high affinity of CC-FLE sE for the natural precursor of OZ-D4. This suggests that the robust viral epitope recognition and neutralization by CC-FLE sE are achievable without extensive somatic mutations, indicating its potential to elicit an immune response within the human B cell repertoire. Consequently, CC-FLE sE emerges as a promising vaccine design strategy for flaviviruses, addressing the dual challenges of immune protection and ADE reduction that are central to flavivirus vaccine development. However, to bolster the rigor and comprehensiveness of the study's conclusions, the authors are advised to address the following key points prior to further consideration:

1) Immunogenicity data: The presented data regarding immunogenicity exhibit inconsistencies and lack persuasiveness. To enhance credibility, it is imperative to conduct PRNT (Plaque Reduction Neutralization Test) titers for Figures 4-6. While Figure 4C illustrates the outcomes of the PRNT, it is crucial to express these findings as either PRNT50 or PRNT90 values for meaningful comparison. Moreover, a statistical correlation analysis between the RVP system results and the PRNT titers must be performed to strengthen the validity of the conclusions drawn. Of particular concern is the discrepancy observed in Figure 5D, where negative outcomes in mortality from passive transfer experiments contradict the reported high levels of neutralizing antibodies. This apparent paradox necessitates a thorough explanation and nuanced discussion within the text.

2) Vaccine efficacy assay: At lines 357-360, the authors' decision to employ Balb/c mice for assessing vaccine efficacy is unconventional and warrants additional justification. Given industry standards, incorporating challenge data from A129 or AG129 mouse models would significantly bolster the study's comprehensiveness. Furthermore, evaluating the immunogenicity and protective efficacy of the vaccine candidate in pregnant mice is highly recommended to address potential concerns related to maternal immunization strategies.

3) Figure 6 depicts solely the effects of adjuvants, which could be more appropriately relocated to the supplementary materials section.

Reviewer #3

(Remarks to the Author)

Wang and colleagues employed a structure-based approach to design the envelope (E) dimer of various flaviviruses using intra- and inter-chain disulfide bonds to improve stability and minimize exposure of the antibody-dependent enhancement (ADE)-prone fusion loop epitope (FLE). The designed constructs were produced as recombinant proteins and their immunogenicity was evaluated in mice, in conjunction with several adjuvants used either alone or in combination. One candidate, CC_FLE sE, derived from the Zika virus (ZIKV) and formulated with an advanced supramolecular adjuvant, emerged as the most potent in inducing a desirable antibody response and providing protection against live ZIKV challenge. The same immunogen also induced monoclonal antibodies targeting the E-dimer-dependent epitope (EDE) in a transgenic mouse model expressing human antibody genes. The authors propose that their designed immunogen, CC_FLE sE, represents a promising strategy for developing ZIKV vaccines that minimize ADE risks while maintaining high protective efficacy. Furthermore, the design strategies may be applicable to vaccines targeting other flaviviruses such as Japanese encephalitis virus (JEV) and tick-borne encephalitis virus (TBEV).

This study is significant. The design of CC_FLE sE is scientifically robust, most of the experiments are well-executed, and the conclusions are largely supported by the data. However, there are several major gaps in understanding the structural, immunological, and protective features of CC_FLE sE, which render some parts of the study less convincing and compelling. Specific questions and suggestions are detailed below:

1. Structural Confirmation: While the antigenicity of CC_FLE sE and CC_Core was evaluated using mAbs specific to EDE, DIII, and FLE, obtaining the crystal structure of CC_FLE sE would confirm that the introduced disulfide bonds are correctly positioned and do not disrupt the overall E-dimer structure.
2. Biophysical Stability: Thermal stability comparisons of CC_FLE sE and CC_Core to the wild-type (WT) protein would provide valuable insights into the enhanced stability conferred by the design.
3. Neutralizing Response (Figure 4): The unexpectedly low neutralizing response induced by CC_Core warrants an explanation. Are there structural or immunological factors that might account for this outcome?
4. Adjuvant Selection (Figure 5B): The neutralizing antibody response induced by CC_FLE with Adjuvax is similar to that with RCEP+R848. Why did the authors prioritize the use of RCEP+R848, which is a complex and mechanistically unclear adjuvant? Additionally, it is surprising that CC_FLE formulated with R848, or CC_FLE alone, failed to induce any detectable neutralizing antibody response.
5. Protective Efficacy (Figure 5D): The absence of protective differences between WT and PBS is puzzling, given the high levels of neutralizing antibodies in WT compared to the absence of detectable antibodies in PBS (Figures 5B and 5D). This discrepancy requires clarification.
6. ADE-Mediated Enhancement (Figures 5E-G): While the results suggest potential ADE-mediated enhancement of DENV-1 and DENV-2 infection in vitro and DENV-2 in vivo, these findings cannot definitively be attributed to FLE-specific antibodies. Competition assays to quantify the proportion of FLE-specific antibodies in the sera and depletion experiments to remove FLE-specific antibodies before conducting in vitro and in vivo studies would strengthen the conclusions.
7. Mouse Model Discrepancies (Figure 6): Substantial differences in neutralizing antibody responses were observed between BALB/c and C57BL/6 mice. Additionally, the authors shifted from using Adjuvax to focusing solely on

RCEP+R848. What is the rationale for this change, and how can the discrepancies observed in C57BL/6 mice be reconciled?

8. Monoclonal Antibody Competition (Figure 7): The isolation and characterization of four mAbs (OZ-D4, B11, A1, and B9) with distinct epitope specificities is interesting. Competing immune sera with these mAbs to estimate the relative proportion of similar antibodies in the sera would be a critical experiment to demonstrate that CC_FLE sE indeed induces higher levels of EDE antibodies compared to WT.

9. ADE Potential of Isolated mAbs (Figure 8): The structure of OZ-D4 indicates it has four-residue contacts within FLE. Investigating whether OZ-D4, along with OZ-B11, A1, and B9, contributes to ADE effects on DENV-1 and DENV-2 in vitro would add valuable insights.

10. Germline Reversion (Figure 9): While the germline reversion experiment is intriguing, the results may be influenced by the specific nature of the transgenic mouse model used. Overstatements regarding these findings should be avoided.

Version 1:

Reviewer comments:

Reviewer #1

(Remarks to the Author)

Reviewer #1 (comment to the authors' responses are in italic)

Comment

1) How the introduced mutations affect the overall structure of the E-dimer? In the E dimer, the N-term of DI, leans towards DIII and the DI-150loop is juxtaposed to the FL. In the TBE dimer, for example, amino acids H5 and G102 form an electrostatic bond only at low pH when the 150-loop is in an "open lid" position away from the FL, but at neutral pH the 150-loop is in a "closed lid" position interacting with G102 while the DI N-term shifts slightly away from FL (doi.org/10.1038/s41467-022-31111-y). The newly created C-C bond between G5 and G102 would force DI N-term to stay close to the FL and affect the 150-loop position and probably also DIII overall conformation. From figure S2A and S2C The CC_FLE dimer looks open in the middle with an altered intradimer interface and a bulgy DIII. The PDB model shows a more closed dimer but the fitting of the model in the density map could be slightly adjusted to assure a more precise reconstruction (within the limits of the relative low resolution).

Response:

CryoEM structures of the ZIKV CC_FLE (new structure, Fig. 2C) and JEV CC_FLE (Fig. 3C, 3D) constructs confirm the integrity of the engineered dimers. The engineered G5C/G102C disulfide bonds stabilizing the dimers are visualized in both structures (Fig2D., Fig3D CC bond with map). In ZIKV CC_FLE structure the 150-loop is disordered. In JEV CC-FLE, the 150-loop adopts very similar configuration (yellow, with glycan 154) as in unmodified JEV dimer (pdb 3p54, blue) and very different from TBEV E dimer structure (pdb 7qre, red) (see Figure below)

Comment:

Since the proposed constructs will be used as antigens it is important to verify not only that the FLE is not exposed but also that the overall structure of the dimer is not altered. The comparison between the ZIKV CC_FLE dimer structure in complex with an antibody and the wt Zika dimer or virus-derived dimers is proposed in table S2. This comparison is presented only as a table and no structure of the ZIKV CC_FLE alone is presented. A figure showing the overlap of the compared structures would be informative.

The 150loop structure (disordered or not) is not mentioned in the text. The comparison of the 150loop (shown here) between the two JEV structures is odd because 3p54 dimer was produced in bacteria so it is not glycosylated. Why was not compared to the JEV dimer from 5wsn?

The conformation of the engineered dimer is crucial to determine the type of antibodies that can be generated, and the data presented in this work do not strictly support the superiority of ZIKV CC_FLE dimer in comparison to the other described constructs.

Comment

2) What is the rationale for the strategy reported in lines 169-175? Has the structure of DI-DIII construct been analyzed? DI may not be correctly folded, and this antigen would focus the response only to DIII and probably also to DIII epitopes that usually are not exposed in the dimer.

Response:

The rationale is to delete DII (residues 51–133 and 195–284) or DI and DII (residues 1–302), which contain more conserved amino acid residues including the FLE that form ADE-prone epitopes, in order to prevent the potential elicitation of ADE-inducing antibodies. Since the immunogenicity outcome of DI-DIII is suboptimal, we therefore did not further investigate the structure.

Comment:

Reduced immunogenicity is probably not the cause of the absence of response for DI-DIII construct because DIII construct seems to be immunogenic. May be the overall structure is misfolded? I would mention this possibility in the text (line 248-249).

Comment

3) Fig.8: The footprint of OZ-D4 antibody does not overlap with the ones of the EDE antibodies. OZ-D4 binds DII and DIII but not DI (differently from the EDEs) and approaches the dimer from the side. This binding would clash with the herringbone arrangement of the dimer on the virus. A docking of this antibody on the cryoEM of the Zika particle would allow to check the accessibility of the E dimer to OZ-D4. Neutralization has been tested with the RVP system but not with authentic virus. It would be interesting to verify the affinity of OZ-D4 for the dimer in the context of the herringbone organization.

Response:

Structural analysis indicates that OZ-D4 approaches the sE dimer in the herringbone arrangement at a more lateral angle (figure on right) (OZ-D4 in red), whereas SMZAb2 (in green) at a more vertical angle. OZ-D4 demonstrated neutralization activity against authentic ZIKV virus (new data, Fig. 7G), as well as the RVP, despite with reduced potency compared to the RVP, suggesting that it can overall overcome the steric hindrance on the virus surface.

Comment:

The sentence at lines 494-497 need to be completed by the notation that OZ-D4 shows a 23-fold reduced capacity of neutralization with the authentic virus compared to the RVP system. This is not the case for C8 (fig.7F-7G) suggesting that OZ-D4 does not behave like an EDE antibody and the binding at a more lateral angle and steric hindrance on the virus surface may be the reason for the observed reduced neutralizing activity. It is interesting that OZ-D4 binding to CC_FLE resulted in low resolution cryoEM and the structure could be obtained only with CC_Core which is more flexible and can probably better adapt to the OZ-D4 binding. These aspects are not discussed.

Comment:

2) Some of the FLE antibodies, like 2A10G6, bind G102. Have the authors considered the possibility that the lack of binding of the CC_FLE antigen could be due to the mutation of this amino acid more than a general non-exposure of the FL?

Response:

Mutation G102C by itself is not sufficient to abolish FLE-mAb binding, as minor monomeric fraction of ZIKV CC_FLE sE could still bind FEL-mAbs, like WT sE. Therefore, the reduced FLE-mAb binding of CC_FLE is caused by the non-exposure of the FL resulting from the G102C/G5C mutations.

Comment:

Can the authors show and comment these data? May be add a supplementary figure?

Comment:

3) Why the binding of FLE antibodies (Fig.2C) is different between WT and CC_Core? The FL should be exposed equally in both constructs because the dimer is "breathing".

Response:

Briefly, The FL in monomeric WT sE is fully exposed, while the CC_Core dimer is "breathing" (T_m increased by only 3 oC compared to WT, Fig. 2B) with FL partially exposed, the CC_FLE dimer is much "locked" with FL heavily occluded.

Comment:

In addition to the increased T_m (due to the presence of the C-C bond), what are the other evidences that the "breathing" dimer "partially exposes the FL"? A similar construct described in DOI: 10.1038/ncomms15411 showed no difference with WT sE for antibody binding or insertion into liposomes.

Comment:

4) The neutralization data reported in Fig.4B are not clear. How C264 mutation would affect antigenicity? The impact of this mutation on the structure does not explain the different response obtained compared to WT construct. Also, DI-DIII and DIII construct bind similarly to DIII antibodies, but the neutralization profile is different. Have these sera been tested against authentic virus? May be the organization of the E protein on the RVP plays a role.

Response:

Antigenicity (antibody binding profile) of a given immunogen (tested in vitro) does not always translate to in vivo immunogenicity (e.g. neutralizing antibody titers), this is often observed in immunogen design studies. Some other factors, such as protein stability, could also play a role in impacting the presentation of neutralizing epitopes. In the case of the different immunogenicity outcome of CC_FLE and CC_Core immunogen, the substantially higher thermostability of CC_FLE may enable this E dimer presenting neutralizing antibody epitopes in the germinal center for a longer duration than the CC_Core. Additionally, the "beathing" dimer nature of CC-Core may also partially affect the presentation of DIII LR epitope that is well exposed in the monomeric sE. This may help to explain the poor immunogenicity of CC_core immunogen after two immunizations. In a different study, after the third immunization, CC_Core immunized animals started to show higher neutralizing antibody titers. Therefore, CC_Core does not improve immunogenicity compared to the WT sE. The DI-DIII configuration may lead to a decreased protein stability due to the deletion of DII, thus having a different immunogenicity outcome from DIII.

Comment:

Has stability been tested in other context (cell culture) than Tm increase that is probably due to the presence of C-C bond?

Comment:

5) Fig.5G: How was the amount of serum to be passively transferred chosen? How would a positive control (Virus + FLE antibody at enhancing concentration) behave in this system?

Response:

The amount of serum to be passively transferred followed the previous publications (PMID: 28340344). We have not used any FLE antibody as control. The WT sE polyclonal sera in our study showed behave very consistently with previously published studies (PMID: 28340344).

Comment:

Publication PMID:28340344 could be cited in the text to justify the amount of serum used.

The virus+FLE antibody control in this context would proof that the reduced enhancing effect of CC_FLE immune sera is due to the non-exposure of FLE.

Comment:

10) It is interesting that the complex of OZ-D4 with CC_FLE did not result in high resolution data while the complex with CC_Core did. The antibody binds the two antigens with comparable affinity (Fig.7D).

Response:

We now have obtained cryoEM structure of SMZAb2 in complex with CC_FLE sE (Fig. 2C).

Comment:

My question was referring to the difference of OZ-D4 epitope exposure in the two constructs. The ELISA data show no difference in binding but the cryoEM data suggest a different (weaker? heterogenous?) binding that affect the resolution of the complex. Could the authors comment on this discrepancy?

Moreover, it would be useful to show in the figures of the different structures, the contact bonds between amino acids and have a table with a list of the polar and salt bridges contacts to better appreciate the footprint of the different antibodies on the dimer. Please show the entire dimer in fig. 8F for all the antibodies.

Additional points:

-Lines 165-166: Has the occlusion of W101 and F108 residues been seen in the cryoEM of CC_FLE dimer? The structure of this dimer for Zika is presented only in complex with antibodies but not alone, so it is difficult to compare the impact of these mutations on the overall structure of the WT E dimer. The close view shown in fig.1B should be compared to the same position in the context of the WT dimer not the monomer (see legend description).

-Fig.S1A: SEC profile per se does not proof that the protein is a dimer in solution. Elution volume is correlated to the protein size for globular proteins, but the envelope protein is elongated, and it has been shown in the literature to elute at unexpected volumes. A MALS analysis or an SDS-PAGE analysis would be more adequate proof.

-Fig.S1B: The legend of BLI graph is not shown (different protein concentrations). Ab Z006 is not shown in the graph, and it can be removed from the table (fig.S1D).

-Fig. 3E (as well S3D and E) need to show an overlap with the unlocked dimer to appreciate the occlusion of the fusion loop.

-Fig. S8: these data do not show superior in vivo protection capacity of the CC_FLE construct in comparison to the WT. There is some inconsistency in the data on adult Balb/c mice at day 1 post challenge showing higher reduction of challenge viral RNA for CC_FLE vs WT in fig.6C and no difference in fig. S8C. Moreover, the challenge viral RNA is equally reduced in the uterus tissue for both constructs (1/8 vs 0/8 is not significant) thus the sentence at lines 437-439 would need a reformulation because the CC_FLE construct does not "outperform" the WT construct.

Reviewer #2

(Remarks to the Author)

The authors have addressed most of the comments raised during the previous round of review. The reviewer suggests that the authors cite the references (PMID: 27603093, PMID: 27855206) that firstly report the use of immunocompetent mice in ZIKV research.

Reviewer #3

(Remarks to the Author)

The authors have satisfactorily addressed all of my concerns.

Version 2:

Reviewer comments:

Reviewer #1

(Remarks to the Author)

The authors have addressed most of my comments. However, I still believe the data do not fully support the immunogenic superiority of the CC_FLE construct for the following reasons: (a) the 150-loop structure appears to be affected, as seen in the comparison of the JEV construct with the JEV E dimer from the virus (yellow vs. purple); (b) the OZ-D4 antibody generated by this antigen binds the dimer in an "unnatural" way that impacts neutralization in authentic virus; and (c) regardless of the viral load in the animal model, the difference between 1/8 and 0/8 is not significant. I appreciate the effort put into complementing and revising the text according to my suggestions and have no further concerns.

Point-by-Point Response to Reviewer's Comments

We appreciate the overall positive and constructive comments from the reviewers. We have performed additional experiments and added more data into the manuscript including (i) cryoEM structure of ZIKV CC_FLE sE complexed with antibody SMZAb2 (EDE1 class), (ii) thermostability characterization of ZIKV CC_FLE sE, and (iii) ZIKV CC_FLE sE efficacy in pregnancy.

After carefully revising our manuscript, with the edited text highlighted in a word file containing tracked changes, we are providing point-by-point response to their comments as below.

Reviewer #1 (Remarks to the Author):

Comment

Wang and colleagues present in this paper the characterization of an engineered dimer of the Zika virus (ZIKV) envelope protein (E). The Flavivirus E dimer has been shown to be target of potent cross-neutralizing dengue and Zika antibodies (E-dimer epitope antibodies – EDE). The dimer also hide the fusion loop epitope (FLE) which is targeted by antibodies that cause antibody-dependent enhancement (ADE) of infection. FLE antibodies are cross-reactive but poorly neutralizing and ADE is a risk factor for severe pathology during Flavivirus secondary infection. EDE antibodies instead are cross-reactive and highly neutralizing, and several strategies have been developed to engineer antigens able to expose EDE epitopes and hide FLE epitopes. Here, the authors have introduced a G5C/G102C mutations (CC_FLE construct) that generate a disulphide bond between the N-term of domain I (DI) of one protomer and the fusion loop (FL) of the opposite protomer resulting in a locked ZIKV E dimer. They compare this construct to an E-monomer and a E-dimer with a single mutation A264C (CC_Core construct). The latter construct has already been reported in the literature (doi: 10.1074/jbc.RA119.007443; doi: 10.1038/s41598-017-01097-5). The three constructs have been tested for antigenicity and protection in an AG129 mouse and in immunocompetent mice model. In addition, the authors used the CC_FLE antigen to immunize Omni-mouse animals (carrying human Ig-loci) from which 4 antibodies have been cloned and one EDE-like humanized antibody (OZ-D4) has been characterized. The engineered dimer alone or in complex with OZ-D4 has been structurally characterized. The authors gathered an impressive amount of data that however need to be thoroughly discussed in comparison to the already published constructs to highlight the added value of the proposed strategy.

Immunological cross-reactivity between Flaviviruses circulating in the same endemic population makes vaccine design challenging because the antigen to be used in a vaccine formulation, in addition to expose protective epitopes, needs to mask enhancing epitopes and avoid uncovering cryptic epitopes. The strategy to engineer stabilized E-dimer for dengue or Zika has been first proposed by Rouvinski et al. in 2017 (doi: 10.1038/ncomms15411) and used for vaccine design and protection studies by several authors (doi.org/10.1016/j.jbc.2022.102079; doi.org/10.1038/s41590-019-0477-z; doi.org/10.1038/s41590-021-00966-6; doi: 10.1074/jbc.RA119.007443).

Response:

We have cited these references in the text.

From the structural point of view there are few points that are not clear:

Comment

- 1) How the introduced mutations affect the overall structure of the E-dimer? In the E dimer, the N-term of DI, leans towards DIII and the DI-150loop is juxtaposed to the FL. In the TBE dimer, for example, amino acids H5 and G102 form an electrostatic bond only at low pH when the 150-loop is in an “open lid” position away from the FL, but at neutral pH the 150-loop is in a “closed lid” position interacting with G102 while the DI N-term shifts slightly away from FL (doi.org/10.1038/s41467-022-31111-y). The newly created C-C bond between G5 and G102 would force DI N-term to stay close to the FL and affect the 150-loop position and probably also DIII overall conformation. From figure S2A and S2C The CC_FLE dimer looks open in the middle with an altered intradimer interface and a bulgy DIII. The PDB model shows a more closed dimer but the fitting of the model in the density map could be slightly adjusted to assure a more precise reconstruction (within the limits of the relative low resolution).

Response:

CryoEM structures of the ZIKV CC_FLE (new structure, Fig. 2C) and JEV CC_FLE (Fig. 3C, 3D) constructs confirm the integrity of the engineered dimers. The engineered G5C/G102C disulfide bonds stabilizing the dimers are visualized in both structures (Fig2D., Fig3D CC bond with map). In ZIKV CC_FLE structure the 150-loop is disordered. In JEV CC-FLE, the 150-loop adopts very similar configuration (yellow, with glycan 154) as in unmodified JEV dimer (pdb 3p54, blue) and very different from TBEV E dimer structure (pdb 7qre, red) (see Figure below)

Comment

How this structure overlaps with the unlocked dimer (for example from the cryoEM structure of the Zika or JEV virus)? In figure S2D the map density is barely visible, a more intense colour would help to appreciate the amino acids fit in the density. X-ray crystallography for both locked dimers would help in defining the impact of the mutations on the overall structure of the dimer. Structure of the dimer in complex with known human antibodies would allow to verify if the protective epitopes are correctly exposed.

Response:

We compared (Table S2) several structural parameters of Zika CC_FLE dimer with several unmodified wild-type (WT) dimers (e.g., PDB: 5LBS) and virus-derived dimers (e.g., PDB: 6CO8). Zika CC_FLE maintained a similar interprotomer distance (24.5 Å) and crossing angle (8.8°) compared to the virus-derived dimer but exhibited a higher buried surface area (BSA: 1689.8 Å², close to average BSA: 1618 Å² for virus-derived dimer) than the WT dimer (average

BSA: 1477 Å²), indicating enhanced stability. Thus, ZIKV CC_FLE retains a native-like structure while providing a stronger dimer presentation, better mimicking the virus-derived dimer (PDB: 6CO8).

For JEV, structural comparisons showed that the wild-type dimer (e.g., PDB: 3p54) had the largest interprotomer distance (26.5 Å) with a smaller BSA (842.7 Å²) (Table S2). The virus-derived dimer (PDB: 5wsn) had a more compact structure with interprotomer distance 25.4 Å, and a higher BSA (1129.1 Å²). The modified JEV_FLE dimer closely resembled the virus-derived dimer, sharing the same interprotomer distance (25.4 Å) but with an intermediate crossing angle (7.4°) and increased BSA (1191.8 Å²). This suggests that JEV_FLE version of the dimer shifts toward a virus-like conformation, enhancing its stability and displaying more natural virus-like antigenic presentation.

We have enhanced the map density of Fig. 2D and Fig. 3D, which is more visible. We have obtained Cryo-EM structure of the ZIKV CC_FLE-sE protein complex with known human SMZAb2 Fab (Fig. 2C, 2D, Fig. S2), which verified that the EDE1 epitope is correctly presented in our modified dimer.

Comment

2) What is the rationale for the strategy reported in lines 169-175? Has the structure of DI-DIII construct been analyzed? DI may not be correctly folded, and this antigen would focus the response only to DIII and probably also to DIII epitopes that usually are not exposed in the dimer.

Response:

The rationale is to delete DII (residues 51–133 and 195–284) or DI and DII (residues 1–302), which contain more conserved amino acid residues including the FLE that form ADE-prone epitopes, in order to prevent the potential elicitation of ADE-inducing antibodies. Since the immunogenicity outcome of DI-DIII is suboptimal, we therefore did not further investigate the structure.

Comment

3) Fig.8: The footprint of OZ-D4 antibody does not overlap with the ones of the EDE antibodies. OZ-D4 binds DII and DIII but not DI (differently from the EDEs) and approaches the dimer from the side. This binding would clash with the herringbone arrangement of the dimer on the virus. A docking of this antibody on the cryoEM of the Zika particle would allow to check the accessibility of the E dimer to OZ-D4. Neutralization has been tested with the RVP system but not with authentic virus. It would be interesting to verify the affinity of OZ-D4 for the dimer in the context of the herringbone organization.

Response:

Structural analysis indicates that OZ-D4 approaches the sE dimer in the herringbone arrangement at a more lateral angle (figure on right) (OZ-D4 in red), whereas SMZAb2 (in green) at a more vertical angle. OZ-D4 demonstrated neutralization activity against authentic ZIKV virus (new data, Fig. 7G), as well as the RVP, despite with reduced potency compared to the RVP, suggesting that it can overall overcome the steric hindrance on the virus surface.

For the antigenicity studies:

Comment:

1) Could the authors clarify the epitope targeted by antibody SMZAb2? Is this the antibody isolated by Magnani et al. (doi: 10.1126/scitranslmed.aan8184)? SMZAb2 in this publication has been classified as an FL antibody, is binding in ELISA to a ZIKV E protein and is competing with FL antibody 4G2. Here the authors classify SMZAb2 as an EDE that does not recognize the WT monomer construct (Fig. 2C). In addition to EDE-C8 have the authors tested EDE-C10? This antibody has shown to be more sensible to the constraints induced in the E-dimer by engineered C-C bonds and could give information about the flexibility of the C5/C102 E-dimer.

Response:

We have obtained cryoEM structure of ZIKV CC_FLE sE complexed with SMZAb2 (Fig. 2C, Fig. S2, Table S1), with epitope analysis reveals that the SMZAb2 epitope significantly overlaps with EDE1-C8, a well-characterized broadly neutralizing antibody (Fig. S2F-G, Fig. 8E). Of the 32 residues engaged by SMZAb2, 27 are shared with the EDE1-C8 epitope (Fig. S2G), underscoring SMZAb2's classification within the EDE1 antibody family.

We have tested the binding of EDE-C10 to our dimeric immunogens by BLI, with an overall affinity of ~100 nM, indicating that all soluble form of ZIKV sE dimers impose flexibility constraints for the C10 epitope presentation.

Comment:

2) Some of the FLE antibodies, like 2A10G6, bind G102. Have the authors considered the possibility that the lack of binding of the CC_FLE antigen could be due to the mutation of this amino acid more than a general non-exposure of the FL?

Response:

Mutation G102C by itself is not sufficient to abolish FLE-mAb binding, as minor monomeric fraction of ZIKV CC_FLE sE could still bind FEL-mAbs, like WT sE. Therefore, the reduced FLE-mAb binding of CC_FLE is caused by the non-exposure of the FL resulting from the G102C/G5C mutations.

Comment:

3) Why the binding of FLE antibodies (Fig.2C) is different between WT and CC_Core? The FL should be exposed equally in both constructs because the dimer is “breathing”.

Response:

Briefly, The FL in monomeric WT sE is fully exposed, while the CC_Core dimer is “breathing” (T_m increased by only 3 °C compared to WT, Fig. 2B) with FL partially exposed, the CC_FLE dimer is much “locked” with FL heavily occluded.

Comment:

4) The neutralization data reported in Fig.4B are not clear. How C264 mutation would affect antigenicity? The impact of this mutation on the structure does not explain the different response obtained compared to WT construct. Also, DI-DIII and DIII construct bind similarly to DIII antibodies, but the neutralization profile is different. Have these sera been tested against authentic virus? May be the organization of the E protein on the RVP plays a role.

Response:

Antigenicity (antibody binding profile) of a given immunogen (tested in vitro) does not always translate to in vivo immunogenicity (e.g. neutralizing antibody titers), this is often observed in immunogen design studies. Some other factors, such as protein stability, could also play a role in impacting the presentation of neutralizing epitopes. In the case of the different immunogenicity outcome of CC_FLE and CC_Core immunogen, the substantially higher thermostability of CC_FLE may enable this E dimer presenting neutralizing antibody epitopes in the germinal center for a longer duration than the CC_Core. Additionally, the “beathing” dimer nature of CC-Core may also partially affect the presentation of DIII LR epitope that is well exposed in the monomeric sE. This may help to explain the poor immunogenicity of CC_core immunogen after two immunizations. In a different study, after the third immunization, CC_Core immunized animals started to show higher neutralizing antibody titers. Therefore, CC_Core does not improve immunogenicity compared to the WT sE.

The DI-DIII configuration may lead to a decreased protein stability due to the deletion of DII, thus having a different immunogenicity outcome from DIII.

Comment:

5) Fig.5G: How was the amount of serum to be passively transferred chosen? How would a positive control (Virus + FLE antibody at enhancing concentration) behave in this system?

Response:

The amount of serum to be passively transferred followed the previous publications (PMID: 28340344). We have not used any FLE antibody as control. The WT sE polyclonal sera in our study showed behave very consistently with previously published studies (PMID: 28340344).

Comment:

6) Fig.S6A: For the depletion experiments graphs, I would add in the legend a column indicating the Abs specificities left (FLE, DIII, EDE). This would facilitate the reading of the graphs.

Response:

We have added the Ab specificity legend column in this figure (now Fig. S7)

Comment:

7) I would move Fig.S6B to main Fig.6 because it shows the landscape of antibody specificities elicited by the different antigens. I would give more information on how to read the graph. My understanding is that a high Y axis value corresponds to a drop of neutralization activity (i.e. the antigen depletion has removed antibodies important for neutralization). It is interesting that the sera obtained from WT immunization show the highest reduction in neutralization when depleted with the CC_FLE and not when depleted with WT. It would suggest that the immunization with WT has induced also some EDE antibodies (may be because the soluble E protein in solution is in monomer-dimer equilibrium and can also expose dimer epitopes?). These antibodies specificities should be analyzed more in detail and discussed.

Response:

We appreciate the reviewer's suggestion. However, the polyclonal antibody response specificity analysis data is linked to Fig.5, which is already quite full. Therefore, we would leave it in the new Fig. S7. We have similar analysis data from the OmniMouse animal shown in Fig. 7B, which well represents the Ab response specificity of CC_FLE sE immunization. In the figure legend, we have listed how the depletion effect was calculated.

The Y-axis indicates the degree of response to antigen depletion, expressed as the percentage decrease compared to the mock condition (no antigen depletion). For WT sE immune sera, all three antigens showed strong depletion effects, similar to the DIII-specific antibody ZV67, suggesting predominant DIII specificity. Since CC_FLE, WT sE, and DIII all present the DIII lateral ridge (LR) epitope well, ZV67 (DIII-specific) showed strong depletion by CC_FLE, similar to the WT sE sera. Notably, a strong response to CC_FLE depletion alone does not imply high EDE specificity. In contrast, EDE1-C8 was only depleted by CC_FLE, but not by DIII or WT sE, confirming its EDE specificity. The CC_FLE immune sera were primarily depleted by CC_FLE, but showed minimal depletion by WT sE and DIII, resembling the behavior of EDE1-C8. We plan to further analyze sera specificity in future studies, for example, using sE mutants with selectively altered epitopes.

Comment:

8) The OmniMouse model is interesting. Why have the authors not performed a comparison between WT and CC_FLE antigens? I see the interest in cloning specific antibodies from these mice, but it would be also informative to look at the B cell repertoire induced by the two types of antigens to appreciate the superiority of the CC_FLE antigen.

Response:

Thanks to the reviewer's suggestion to compare the WT vs. CC_FLE sE antibody response in the OmniMouse animal model. Since such investigation has been performed in wildtype mouse models, we then focus on isolating the dimer-specific neutralizing antibodies elicited by the CC_FLE sE reported in the current manuscript. We will perform such investigation in Omnimouse animals in follow-up studies in the future.

Comment:

9) Why was the OmniMouse 5.2 terminated at 6 weeks p.i., much earlier than the OmniMouse 5.1 (38 weeks)? From the shape of the neutralization curves, the sera do not show a strong neutralizing activity. Have the sera been tested for presence of FLE antibodies?

Response:

OmniMouse 5.2 was terminated at 6 weeks post-immunization (p.i.), as its serum neutralizing titer at 5 weeks p.i. was 594—above the preset cutoff of 500 for termination and antibody isolation. OmniMouse 5.1 continued to receive additional immunizations, as its serum neutralizing titer at 5 weeks p.i. was below 500. We have mapped the neutralizing antibody specificity to epitopes including EDE and DIII. Due to limited serum volume, we will assess the presence of FLE-directed antibodies in a follow-up study using a larger number of animals and WT sE as a control immunogen to support this investigation.

Comment:

10) It is interesting that the complex of OZ-D4 with CC_FLE did not result in high resolution

data while the complex with CC_Core did. The antibody binds the two antigens with comparable affinity (Fig.7D).

Response:

We now have obtained cryoEM structure of SMZAb2 in complex with CC_FLE sE (Fig. 2C).

Minor points:

Line 173: What is (32-34) referring to? The reference number does not match with “DIII immunogenicity” described in the text.

Response: *We have removed the corresponding text.*

Lines 179-181: It would be informative to show the SEC of profile of the proteins after affinity purification (may be as a supplementary figure) to appreciate the heterogeneity of the forms produced by the constructs.

Response: *We have added the SEC profile after affinity purification in Fig. S1A*

Lines 191-192: Please list all the antibodies that have been tested with their epitope specificity and recognition of the antigens tested. A figure and a summary table in supplementary data would be informative.

Response: *We have generated a Table S5 for this purpose.*

Lines 196-199: This sentence is redundant. These results have been presented in the previous paragraphs.

Response: *We have removed this sentence.*

Fig.5: B and C show neutralization assays results (RVP and authentic virus). I would plot both in the same way (either ID50 or % of neutralization).

Response: *We have listed ID50 values for each pooled serum sample in Fig. 5C.*

Line 364: the ID50 titer increase is not 2 logs (5012 vs 178 = 28 folds).

Response: *We have made the correction as suggested.*

Lines 399-400: I do not understand the sentence “surpassing the nascent WT sE”.

Response: *It means “CC_FLE sE is better than the unmodified WT sE”.*

Fig.8: 8A – show the HC and LC chains with more contrasting colors. 8F – show a top view of the dimer to highlight the antibody footprint. The actual side view is not informative.

Response: *We have modified Fig. 8A to show the surface rendering of the electron density colored based on the Fab chains and antigen regions. Fig. 8F is now shown as a top view of the dimer to illustrate the antibody footprint.*

Reviewer #2 (Remarks to the Author):

The present manuscript by Wang et al presents a novel vaccine design strategy targeting the E protein of the Zika virus (ZIKV), aimed at bolstering immune responses and mitigating the risk of Antibody-dependent enhancement (ADE). The authors have engineered two mutations, G5C/G102C, into the flavivirus E protein, resulting in the formation of inter-chain disulfide

bonds and the creation of E-dimers. The E-dimer formation not only augments the immunogenicity of the vaccine candidate but also masks the ADE-associated site FLE, thereby diminishing the likelihood of ADE. The vaccine candidate, CC-FLE sE, is primarily constructed based on the ZIKV E protein sequence. Upon immunization in mice, it elicits potent neutralizing antibodies that confer protection against ZIKV infection and attenuate the risk of ADE following infections with various flaviviruses. Furthermore, the article delineates the structural and functional analysis of the E-dimer-dependent epitope (EDE) monoclonal antibody OZ-D4, highlighting the high affinity of CC-FLE sE for the natural precursor of OZ-D4. This suggests that the robust viral epitope recognition and neutralization by CC-FLE sE are achievable without extensive somatic mutations, indicating its potential to elicit an immune response within the human B cell repertoire. Consequently, CC-FLE sE emerges as a promising vaccine design strategy for flaviviruses, addressing the dual challenges of immune protection and ADE reduction that are central to flavivirus vaccine development. However, to bolster the rigor and comprehensiveness of the study's conclusions, the authors are advised to address the following key points prior to further consideration:

1) Immunogenicity data: The presented data regarding immunogenicity exhibit inconsistencies and lack persuasiveness. To enhance credibility, it is imperative to conduct PRNT (Plaque Reduction Neutralization Test) titers for Figures 4-6. Moreover, a statistical correlation analysis between the RVP system results and the PRNT titers must be performed to strengthen the validity of the conclusions drawn.

*Response: Thanks to the reviewer for suggesting validation of the RVP method, which has been performed very carefully in a previous study (PMID: 27481466). Shown below is a statistical correlation analysis from this study between the RVP system results and the PRNT titers. Comparison of the mean EC50 for all serum samples evaluated **with both RVPs and infectious virus revealed remarkable agreement.***

[redacted]

Furthermore, we have performed PRNT in our immune sera analysis (Fig. 5C) with PRNT titers listed for comparison, which is consistent with our RVP neutralization data. Moreover, the RVP assay system has been widely used in the field to characterize clinical trial immune response as a standard method currently. To name a few, VRC/NIH used the RVP assay to measure neutralization in two clinical trials, and Takeda uses the same assay for its ZIKV and DENV vaccine studies. Therefore, our data using a well-established and validated high-through-out RVP assay to evaluate immune sera neutralizing titers are solid.

Comment: While Figure 4C illustrates the outcomes of the PRNT, it is crucial to express these findings as either PRNT50 or PRNT90 values for meaningful comparison.

Response: *We have listed the PRNT50 values of the immune sera for comparison in the revised Fig. 5C.*

Comment:

Of particular concern is the discrepancy observed in Figure 5D, where negative outcomes in mortality from passive transfer experiments contradict the reported high levels of neutralizing antibodies. This apparent paradox necessitates a thorough explanation and nuanced discussion within the text.

Response: *Thanks to the reviewer who suggests clarification. We have added the following text to explain such observation.*

“Although the nAb titers of immune sera from animals inoculated with WT sE + Adjuvex , CC_FLE sE+ Adjuvex, or CC_FLE sE +PCEP-R848 were similar, the better protection efficacy conferred by the sera from the mice immunized with CC_FLE sE +PCEP-R848 indicated that other protection mechanisms, such as antibody effector functions including antibody dependent cytotoxicity (ADCC) may play a role in this AG129 mouse model.”

Comment:

2) Vaccine efficacy assay: At lines 357-360, the authors' decision to employ Balb/c mice for assessing vaccine efficacy is unconventional and warrants additional justification. Given industry standards, incorporating challenge data from A129 or AG129 mouse models would significantly bolster the study's comprehensiveness.

Response: *We chose BALB/c mice as the model animal for vaccine efficacy here based on our in-house observation that after ZIKV challenge, BALB/c mice in general develop higher titers of viremia than C57BL/6 mice (used in our earlier immunogenicity studies), suggesting BALB/c mouse as a suitable model animal for evaluating vaccine efficacy. In the current study, we have used various mouse strains to evaluate immunogenicity and efficacy, including AG129 (immunocompromised), C57BL/6 (immunocompetent), and BALB/c (immunocompetent) mice, in order to avoid bias caused by one strain. Our vaccine candidate, ZIKV CC_FLE sE demonstrated superior performance over the unmodified WT sE in all tested mouse strains. We firmly believe that using an immunocompetent mouse model (e.g., BALB/c mouse model) to evaluate protection efficacy would closely mimic the ZIKV natural infection.*

Comment:

Furthermore, evaluating the immunogenicity and protective efficacy of the vaccine candidate in pregnant mice is highly recommended to address potential concerns related to maternal immunization strategies.

Response: We have evaluated the immunogenicity and protective efficacy of the vaccine candidate in pregnant mice, reported in Fig. S8. These results demonstrate that CC_FLE sE provided effective protection against ZIKV in both pregnant mice and their fetuses, and outperformed WT sE in this immunocompetent mouse pregnancy-protection model.

3) Figure 6 depicts solely the effects of adjuvants, which could be more appropriately relocated to the supplementary materials section.

Response: We respectively emphasis that Fig. 6 is important to show the critical effect of the adjuvant, and the protection corelates as a main figure.

Reviewer #3 (Remarks to the Author):

Comment:

Wang and colleagues employed a structure-based approach to design the envelope (E) dimer of various flaviviruses using intra- and inter-chain disulfide bonds to improve stability and minimize exposure of the antibody-dependent enhancement (ADE)-prone fusion loop epitope (FLE). The designed constructs were produced as recombinant proteins and their immunogenicity was evaluated in mice, in conjunction with several adjuvants used either alone or in combination. One candidate, CC_FLE sE, derived from the Zika virus (ZIKV) and formulated with an advanced supramolecular adjuvant, emerged as the most potent in inducing a desirable antibody response and providing protection against live ZIKV challenge. The same immunogen also induced monoclonal antibodies targeting the E-dimer-dependent epitope (EDE) in a transgenic mouse model expressing human antibody genes. The authors propose that their designed immunogen, CC_FLE sE, represents a promising strategy for developing ZIKV vaccines that minimize ADE risks while maintaining high protective efficacy. Furthermore, the design strategies may be applicable to vaccines targeting other flaviviruses such as Japanese encephalitis virus (JEV) and tick-borne encephalitis virus (TBEV).

This study is significant. The design of CC_FLE sE is scientifically robust, most of the experiments are well-executed, and the conclusions are largely supported by the data. However, there are several major gaps in understanding the structural, immunological, and protective features of CC_FLE sE, which render some parts of the study less convincing and compelling. Specific questions and suggestions are detailed below:

Response: We appreciate the reviewer's overall enthusiasm and have made modifications accordingly.

Comment:

1. Structural Confirmation: While the antigenicity of CC_FLE sE and CC_Core was evaluated using mAbs specific to EDE, DIII, and FLE, obtaining the crystal structure of CC_FLE sE would confirm that the introduced disulfide bonds are correctly positioned and do not disrupt the overall E-dimer structure.

Response: We have obtained cryoEM structure of the CC_FLE sE complexed with EDE antibody SMZAb2 that shows the formation of the engineered interchain disulfide bond and the overall E-dimer structure similar to the virus-derived dimers (Fig. 2C, 2D, Fig. S2, Table S1, Table S2).

Comment:

2. Biophysical Stability: Thermal stability comparisons of CC_FLE sE and CC_Core to the wild-type (WT) protein would provide valuable insights into the enhanced stability conferred by the design.

Response: We have determined the thermostability of the mutant sEs, in comparison with the WT sE (Fig. 2B). We have added the following text to report the data:

“Differential scanning calorimetry profiles demonstrate substantially improved thermostability of CC_FLE sE compared to WT sE, with the thermal transition temperature (T_m) increased by 15°C (from 43.5°C to 58.7°C) (Fig. 2B). In contrast, CC_Core sE showed a moderate (3°C) increase in T_m compared to WT sE (Fig. 2B).”

Comment:

3. Neutralizing Response (Figure 4): The unexpectedly low neutralizing response induced by CC_Core warrants an explanation. Are there structural or immunological factors that might account for this outcome?

Response: Antigenicity (antibody binding profile) of a given immunogen (tested in vitro) does not always translate to in vivo immunogenicity (e.g. neutralizing antibody titers), this is often observed in immunogen design studies. Some other factors, such as protein stability, could also play a role in impacting the presentation of neutralizing epitopes. In the case of the different immunogenicity outcome of CC_FLE and CC_Core immunogen, the substantially higher thermostability of CC_FLE may enable this E dimer presenting neutralizing antibody epitopes in the germinal center of B cells for a longer duration than the CC_Core. Additionally, the “beathing” dimer nature of CC-Core may also partially affect the presentation of DIII LR epitope that is well exposed in the monomeric WT sE. This may help to explain the poor immunogenicity of CC_core immunogen after two immunizations. In a different study, after the third immunization, CC_Core immunized animals started to show increased neutralizing antibody titers. Therefore, CC_Core does not improve immunogenicity compared to the WT sE.

Comment:

4. Adjuvant Selection (Figure 5B): The neutralizing antibody response induced by CC_FLE with Adjuplex is similar to that with RCEP+R848. Why did the authors prioritize the use of RCEP+R848, which is a complex and mechanistically unclear adjuvant?

Response: The selection of an adjuvant was based on its higher potency demonstrated in vivo (Fig. 5D, only immune sera from CC_FLE in PCEP+R848 provided protection statistically significant compared with the immune sera from PBS-injected mice, while protection of sera from CC_FLE in Adjuplex group was not statistically significant).

The following additional reasons justify prioritizing the use of PCEP+R848: (1) PCEP is a chemically well-defined molecular adjuvant and vaccine delivery vehicle of nano-dimensions, which is fully water-soluble and forms clear solutions when complexed with R848 and antigens. As a molecular solution, it can be analyzed by a variety of physico-chemical techniques, such as HPLC, DLS, AF4, UV-spectroscopy, etc. In contrast, Adjuplex is a multi-component bi-phasic adjuvant, which consists of a carbomer (polyacrylate) and soybean lecithin formulated as submicron-sized liposomes as main components (doi:10.1128/CVI.00736-14). The formulation and characterization of PCEP+R848 are much more detailed and thoroughly conducted.

(2) In contrast to Adjuvex, which is based on polyacrylate with a hydrolytically and biologically stable carbon-carbon backbone, PCEP is fully biodegradable. Furthermore, PCEP is a structural homologue of another polyphosphazene adjuvant – PCPP, which showed efficacy and excellent safety profile in five clinical trials (please see summary on degradability of PCEP, its mechanism of action and clinical use of polyphosphazene adjuvants in reference 49 – Andrianov, Langer, *J Control Release* 329, 299-315, doi:10.1016/j.jconrel.2020.12.001 (2021)). In contrast, detailed reports on clinical trials of Adjuvex were not available.

(3). Adjuvex has been commercially unavailable for several years lately.

Comment: Additionally, it is surprising that CC_FLE formulated with R848, or CC_FLE alone, failed to induce any detectable neutralizing antibody response.

Response: *The lack of adjuvant effect of R848 in the absence of an appropriate delivery vehicle (e.g., R848 alone) is well-established. The main reasons for that are the lack of association of this small molecule with the antigen and its rapid clearance from the body (doi: 10.1586/14760584.6.5.835). These deficiencies are alleviated by the complexation of R848 with PCEP in the form of a counterion.*

Adjuvants are a critical component of vaccines, complementing immunogens to enhance and shape immune responses. They can influence the affinity, specificity, magnitude, and functional profile of both B and T cell responses. It is commonly observed that antigen alone often fails to elicit a detectable neutralizing antibody response—for example, as seen with CC_FLE alone in our study. In another case (PMID: 33422991, Fig. 2C), the SARS-CoV-2 spike protein alone induced negligible neutralizing responses in mice, whereas the inclusion of various adjuvants led to robust neutralizing antibody titers.

Comment:

5. Protective Efficacy (Figure 5D): The absence of protective differences between WT and PBS is puzzling, given the high levels of neutralizing antibodies in WT compared to the absence of detectable antibodies in PBS (Figures 5B and 5D). This discrepancy requires clarification.

Response: *We have added the following text to explain the observation: “Although the nAb titers of immune sera from animals inoculated with WT sE + Adjuvex, CC_FLE sE+ Adjuvex, or CC_FLE sE +PCEP-R848 were similar, the better protection efficacy conferred by the sera from the mice immunized with CC_FLE sE +PCEP-R848 indicated that other protection mechanisms, such as antibody effector functions including antibody dependent cytotoxicity (ADCC) may play a role in this AG129 mouse model.”*

Comment:

6. ADE-Mediated Enhancement (Figures 5E-G): While the results suggest potential ADE-mediated enhancement of DENV-1 and DENV-2 infection in vitro and DENV-2 in vivo, these findings cannot definitively be attributed to FLE-specific antibodies. Competition assays to quantify the proportion of FLE-specific antibodies in the sera and depletion experiments to remove FLE-specific antibodies before conducting in vitro and in vivo studies would strengthen the conclusions.

Response: *We appreciate the reviewer’s suggestion to define that the FLE-specific antibody contributes to the WT sE immune sera ADE effect. We envision that CC-FLE may abolish the*

elicitation of FLE-specific ADE-prone antibodies, but also for antibodies with other ADE-epitopes elsewhere including the N-terminus of sE, caused by the “locked” dimeric configuration. We are going to map the ADE-specificity thoroughly in follow-up studies.

We have added the following text in the discussion section to elaborate this point “This engineered linkage further stabilizes the sE dimer shown by substantially increased T_m by 15°C (Fig. 2B) and occludes the FLE (Fig. 2B-C), as well as the sE N-terminus, which is adjacent to another ADE epitope consisting residues N8-E13 of DENV sE —identical to those in the corresponding regions of ZIKV sE. Future studies aimed at delineating the effects of the G5C/G102C mutations on various ADE epitopes would be informative.”

Comment:

7. Mouse Model Discrepancies (Figure 6): Substantial differences in neutralizing antibody responses were observed between BALB/c and C57BL/6 mice.

Response: We chose BALB/c mice as the model animal for vaccine efficacy here based on our in-house observation that after ZIKV challenge, BALB/c mice in general develop higher titers of viremia than C57BL/6 mice (used in our earlier immunogenicity studies), suggesting BALB/c mouse as a suitable model animal for evaluating vaccine efficacy. It is very common that a given immunogen can induce different immune responses in different mouse strains. However, in C57BL/6 (two studies, Fig. 4, Fig. 5) or BALB/c mice (two studies, Fig. 6, Fig. S8), CC_FLE sE was able to consistently elicit potent neutralizing antibody responses. In contrast, the performance of WT sE varies (two studies in C57BL/6 Fig. 4, Fig. 5 and BALB/c, Fig. 6, Fig. S8, respectively). This demonstrates that CC_FLE sE has immunogenicity superior to WT sE, in all tested mouse strains.

Comment:

Additionally, the authors shifted from using Adjuvax to focusing solely on RCEP+R848. What is the rationale for this change, and how can the discrepancies observed in C57BL/6 mice be reconciled?

Response: The rationale of shifting from Adjuvax to PCEP+R848 is based on (i) Adjuvax became commercially unavailable recently, and (ii) the supporting data in Fig. 5D (survival curve) and Fig. S6B (health score) that showed sera from CC_FLE sE in PCEP+R848 behaved better than CC-FLE sE in Adjuvax. AG129 mice receiving sera from immune mice inoculated with CC_FLE formulated with PCEP+R848 adjuvant displayed significantly longer survival time (median, 15 days) upon challenge than from animals inoculated with PBS (median, 11 days) ($p < 0.05$,) or WT sE formulated with Adjuvax (median, 11 days) (Fig. 5D). AG129 mice receiving sera from donor mice immunized with CC_FLE formulated with Adjuvax also showed median survival time of 13 days (Fig. 5D), longer than PBS- or WT sE + Adjuvax inoculated group, although this is not statistically significant. AG129 mice receiving immune sera from CC_FLE and PCEP-R848 adjuvant-immunized animals also showed lower health score (multiple t test, *** $p < 0.001$, days 11 and 13 post infection, Fig. S6B) than from PBS inoculated animals. In contrast, AG129 mice receiving immune sera from CC_FLE in Adjuvax did not show lower health scores statistically significant than the PBS inoculated animals.*

Regarding the discrepancies observed in C57BL/6 mice, as discussed earlier, a given immunogen can elicit different immune responses across mouse strains. Although the mechanism

is not fully understood, we speculate that the threshold for sE to induce a neutralizing antibody response is higher in BALB/c mice than in C57BL/6 mice. While CC_FLE sE appears able to overcome this threshold in both strains, WT sE may struggle to do so in BALB/c mice. Consistently, CC_FLE sE elicited potent neutralizing antibody responses in both C57BL/6 and BALB/c mice, whereas WT sE showed variable performance for unknown reasons. These findings suggest that CC_FLE sE exhibits more consistent and superior immunogenicity compared to WT sE across the tested mouse strains.

Comment:

8. Monoclonal Antibody Competition (Figure 7): The isolation and characterization of four mAbs (OZ-D4, B11, A1, and B9) with distinct epitope specificities is interesting. Competing immune sera with these mAbs to estimate the relative proportion of similar antibodies in the sera would be a critical experiment to demonstrate that CC_FLE sE indeed induces higher levels of EDE antibodies compared to WT.

Response: In wild-type mice (Fig. S7), we provided evidence that sera from CC_FLE sE-immunized mice exhibited neutralization specificity resembling both EDE1-C8-like and ZV67-like (DIII LR) epitopes, whereas sera from WT sE-immunized mice showed primarily ZV67-like (DIII LR) specificity. In Omnimouse animals (Fig. 7B, Fig. S9C), we demonstrated that the neutralizing activity of CC_FLE sE-immunized sera—similar to that of EDE1-C8—could be efficiently depleted by CC_FLE sE, but not so by the WT sE antigen. This suggests that CC_FLE sE effectively elicits antibodies targeting the EDE epitope. As suggested by the reviewer, we will pursue follow-up studies in Omnimouse animals using larger cohort sizes, with WT sE-immunized mice included as controls to enable a more rigorous comparison.

Comment:

9. ADE Potential of Isolated mAbs (Figure 8): The structure of OZ-D4 indicates it has four-residue contacts within FLE. Investigating whether OZ-D4, along with OZ-B11, A1, and B9, contributes to ADE effects on DENV-1 and DENV-2 in vitro would add valuable insights.

Response: None of these antibodies bind DENV-1 or DENV-2 sE. Therefore, we conclude that they have no ADE effect on DENV.

10. Germline Reversion (Figure 9): While the germline reversion experiment is intriguing, the results may be influenced by the specific nature of the transgenic mouse model used. Overstatements regarding these findings should be avoided.

Response: We appreciate the reviewer's comment. The germline sequence is directly derived from the human Ig germline database, not from the transgenic mouse. Of course, further clinical trial data would provide more direct evidence. We have edited related text in the discussion section accordingly.

Point-by-Point Response to Reviewer's Comments

We appreciate the positive and constructive comments from reviewers. We have performed additional experiments and added more data into the manuscript and revised certain figures and related text.

After carefully revising our manuscript, with the edited text highlighted in a word file containing tracked changes, we are providing point-by-point response to their comments as below.

Reviewer #1 (previous comment in grey, current comment to the authors' responses are in italic)

Comment

1) How the introduced mutations affect the overall structure of the E-dimer? In the E dimer, the N-term of DI, leans towards DIII and the DI-150loop is juxtaposed to the FL. In the TBE dimer, for example, amino acids H5 and G102 form an electrostatic bond only at low pH when the 150-loop is in an "open lid" position away from the FL, but at neutral pH the 150-loop is in a "closed lid" position interacting with G102 while the DI N-term shifts slightly away from FL (doi.org/10.1038/s41467-022-31111-y). The newly created C-C bond between G5 and G102 would force DI N-term to stay close to the FL and affect the 150-loop position and probably also DIII overall conformation. From figure S2A and S2C The CC_FLE dimer looks open in the middle with an altered intradimer interface and a bulgy DIII. The PDB model shows a more closed dimer but the fitting of the model in the density map could be slightly adjusted to assure a more precise reconstruction (within the limits of the relative low resolution).

Response:

CryoEM structures of the ZIKV CC_FLE (new structure, Fig. 2C) and JEV CC_FLE (Fig. 3C, 3D) constructs confirm the integrity of the engineered dimers. The engineered G5C/G102C disulfide bonds stabilizing the dimers are visualized in both structures (Fig2D., Fig3D CC bond with map). In ZIKV CC_FLE structure the 150-loop is disordered. In JEV CC-FLE, the 150-loop adopts very similar configuration (yellow, with glycan 154) as in unmodified JEV dimer (pdb 3p54, blue) and very different from TBEV E dimer structure (pdb 7qre, red) (see Figure below)

Comment:

Since the proposed constructs will be used as antigens it is important to verify not only that the FLE is not exposed but also that the overall structure of the dimer is not altered. The comparison between the ZIKV CC_FLE dimer structure in complex with an antibody and the wt Zika dimer or virus-derived dimers is proposed in table S2. This comparison is presented only as a table and no structure of the ZIKV CC_FLE alone is presented. A figure showing the overlap of the compared structures would be informative.

Response: We made an additional supplementary figure, **Fig. S2H**, showing the comparison between the ZIKV CC_FLE dimer structure and the WT ZIKV dimer. These two structures are overall very similar.

Comment:

The 150loop structure (disordered or not) is not mentioned in the text. The comparison of the 150loop (shown here) between the two JEV structures is odd because 3p54 dimer was produced in bacteria so it is not glycosylated. Why was not compared to the JEV dimer from 5wsn?

Response: We have added in the revised text that the 150-loop is disordered in ZIKV CC_FLE structure. **[Lines 194-197]**

We have added 5wsn structure (purple) for comparison of 150-loop conformations. In ZIKV CC_FLE structure, the 150-loop is disordered. In JEV CC-FLE, the 150-loop adopts very similar configuration (yellow, with glycan 154) as in both unmodified JEV E proteins (pdb 3p54, blue; and pdb 5wsn, purple) and very different from TBEV E dimer structure (pdb 7qre, red) (see Figure below).

Comment:

The conformation of the engineered dimer is crucial to determine the type of antibodies that can be generated, and the data presented in this work do not strictly support the superiority of ZIKV CC_FLE dimer in comparison to the other described constructs.

Response: Structural and functional analyses of our CC_FLE construct suggest that it closely mimics the native conformation of the E protein. Our in vivo immunogenicity and virus challenge studies demonstrated that after only two immunizations CC_FLE confers protective efficacy against ZIKV challenge. Notably, in pregnant mice, those immunized with CC_FLE had virus-free uteri and fetus, in contrast to mice immunized with other constructs published previously, which often require more than two immunizations and showed high frequency of uterine and fetus viral infection.

2) Comment:

Reduced immunogenicity is probably not the cause of the absence of response for DI-DIII construct because DIII construct seems to be immunogenic. May be the overall structure is misfolded? I would mention this possibility in the text (line 248-249).

Response: we have mentioned this possibility in the revised text. **[Line 260]**

3) Comment:

The sentence at lines 494-497 need to be completed by the notation that OZ-D4 shows a 23-fold reduced capacity of neutralization with the authentic virus compared to the RVP system. This is not the case for C8 (fig.7F-7G) suggesting that OZ-D4 does not behave like an EDE antibody and the binding at a more lateral angle and steric hindrance on the virus surface may be the reason for the observed reduced neutralizing activity. It is interesting that OZ-D4 binding to CC_FLE resulted in low resolution cryoEM and the structure could be obtained only with CC_Core which is more flexible and can probably better adapt to the OZ-D4 binding. These aspects are not discussed.

Response: we have discussed the approaching angle and footprint difference part. **[Lines 547-554]**

Regarding why OZ-D4 binding to CC_FLE resulted in low-resolution cryoEM data, and why the structure could only be obtained with the more flexible CC_Core, we offer the following explanation. While, in general, more rigid and/or stable protein molecules and complexes are expected to yield higher-resolution structures, this correlation is not always straightforward—particularly in cryoEM. Many other factors influence resolution, with the orientational distribution

of observed particles being one of the most critical, especially as it relates to the ice properties required for high-quality data collection. This is a complex issue, and it is generally not possible to assert that a more rigid protein complex will necessarily produce higher-resolution data. Nevertheless, after testing several antibody–antigen combinations, we successfully obtained the high-resolution structure of neutralizing antibody OZ-D4 complexed with the CC_Core dimer, which clearly reveals the epitope in detail.

2) Comment:

Can the authors show and comment these data? May be add a supplementary figure? [Mutation G102C by itself is not sufficient to abolish FLE-mAb binding, as minor monomeric fraction of ZIKV CC_FLE sE could still bind FEL-mAbs, like WT sE. Therefore, the reduced FLE-mAb binding of CC_FLE is caused by the non-exposure of the FL resulting from the G102C/G5C mutations.]

Response: We have added a new panel as **Fig. S1C**, showing the monomeric fraction of CC_FLE binds FLE-specific mAb E60 by ELISA. **[Line 222-226]**

3) Comment:

In addition to the increased T_m (due to the presence of the C-C bond), what are the other evidences that the “breathing” dimer “partially exposes the FL”? A similar construct described in DOI: 10.1038/ncomms15411 showed no difference with WT sE for antibody binding or insertion into liposomes.

Response: Thanks to the review for sharing the article link. We have similar observations here. In **Fig. 2F**, we have shown that the “breathing” dimer CC_Core shows moderate binding to FLE-specific antibodies, which suggests the “breathing” nature of this type of dimer. In contrast, the locked dimer CC_FLE shows abolished FLE-specific antibody binding.

4) Comment:

Has stability been tested in other context (cell culture) than T_m increase that is probably due to the presence of C-C bond?

Response: We have not assessed stability in the context of cell culture. This is an interesting idea that could be explored in future studies. Nevertheless, we believe that the T_m measurements obtained with purified protein are predictive of stability in other contexts.

5) Comment:

Publication PMID:28340344 could be cited in the text to justify the amount of serum used. The virus+FLE antibody control in this context would proof that the reduced enhancing effect of CC_FLE immune sera is due to the non-exposure of FLE.

Response: We have cited_PMIID:28340344 as reference for the volume of the transferred sera **[Lines 806-807]**.

Our data confirms that immunization with our CC_FLE mitigates ADE risk in a mouse model in addition to the in vitro cell-based ADE assay. The ADE mitigation could be caused by the reduced exposure of FLE, but other factors may contribute to this property as well. For example, CC_FLE mutations may reduce the exposure of other ADE-prone epitopes yet to be identified. We will consider adding virus+FLE antibody control in future studies.

10) Comment:

My question was referring to the difference of OZ-D4 epitope exposure in the two constructs.

The ELISA data show no difference in binding but the cryoEM data suggest a different (weaker? heterogenous?) binding that affect the resolution of the complex. Could the authors comment on this discrepancy?

Moreover, it would be useful to show in the figures of the different structures, the contact bonds between amino acids and have a table with a list of the polar and salt bridges contacts to better appreciate the footprint of the different antibodies on the dimer. Please show the entire dimer in fig. 8F for all the antibodies.

Response: We don't observe significant difference for OZ-D4 epitope exposure in the two constructs. As discussed earlier, the different resolution of structure could be resulted from other factors which may influence resolution, with the orientational distribution of observed particles being one of the most critical, especially as it relates to the ice properties required for high-quality data collection. **[Lines 516-518]**

We generated a new supplementary **TableS6** to summarize the interactions. **[Lines 541-543]**

We have generated new figure panels (Fig. 8G & 8H) to show dimer approaching angle comparison: OZ-D4: CC_Core complex, EDE1-C8: WT Dimer complex (PDB:5LBS), and SMZAb2: CC_FLE complex. **[Lines547-554]**

Please note we have already shown the entire dimer in **Fig. 8F**.

Additional points:

-Lines 165-166: Has the occlusion of W101 and F108 residues been seen in the cryoEM of CC_FLE dimer? The structure of this dimer for Zika is presented only in complex with antibodies but not alone, so it is difficult to compare the impact of these mutations on the overall structure of the WT E dimer. The close view shown in fig.1B should be compared to the same position in the context of the WT dimer not the monomer (see legend description).

Response: The occlusion of W101 and F108 residues has been demonstrated in the cryoEM of CC_FLE dimer **[Lines 205-207]**. The close view shown in **Fig.1B** is just a model based on the WT dimer. The purpose is to explain the rationale of design, not to compare the structure.

We have added **Fig. S2I**, showing the occlusion of W101 and F108 in the CC_FLE dimer. **[Lines 205-207]**

-Fig.S1A: SEC profile per se does not proof that the protein is a dimer in solution. Elution volume is correlated to the protein size for globular proteins, but the envelope protein is elongated, and it has been shown in the literature to elute at unexpected volumes. A MALS analysis or an SDS-PAGE analysis would be more adequate proof.

Response: We have added an SDS-PAGE gel figure (**Fig. S1B**) showing that, under non-reducing conditions, the SEC-purified CC_FLE or CC_Core sE migrated at a molecular weight consistent with a dimer. More decisively, our structural analysis confirmed the dimeric nature of our constructs. **[Line 185]**

-Fig.S1B: The legend of BLI graph is not shown (different protein concentrations). Ab Z006 is not shown in the graph, and it can be removed from the table (fig.S1D).

Response: legend for protein concentration in the BLI graph is now provided (**currently Fig. S1D**). Ab Z006 is omitted from the table (**currently Fig. S1F**).

-Fig. 3E (as well S3D and E) need to show an overlap with the unlocked dimer to appreciate the occlusion of the fusion loop.

Response: **Fig. 3E, Fig. S3D & S3E** are revised as suggested.

-Fig. S8: these data do not show superior in vivo protection capacity of the CC_FLE construct in comparison to the WT. There is some inconsistency in the data on adult Balb/c mice at day 1 post challenge showing higher reduction of challenge viral RNA for CC_FLE vs WT in fig.6C and no difference in fig. S8C. Moreover, the challenge viral RNA is equally reduced in the uterus tissue for both constructs (1/8 vs 0/8 is not significant) thus the sentence at lines 437-439 would need a reformulation because the CC_FLE construct does not “outperform” the WT construct.

Response: The discrepancy between **Fig. 6C** and **Fig. S8C** was due to the different virus challenge routes. In **Fig. 6C**, the virus was administered via the subcutaneous route, which mimics the natural transmission of ZIKV by mosquito bites. The negative control mice showed serum virus loads below 10^4 viral genome copies/mL. As expected, CC_FLE sE showed better efficacy than WT sE. In **Fig. S8C**, we used the retro-orbital route that directly injected virus into the blood to effectively infect the female mouse uterus, which resulted in much higher viral loads, with negative control mice showing levels above 10^5 viral genome copies/mL. The effective ZIKV infection in vivo was therefore about 10-fold higher in Fig. S8C than in **Fig. 6C**, which explains why we did not observe a statistically significant reduction in viral load between WT and CC_FLE sE-immunized mice under such overwhelming viral infection condition. Nevertheless, none of the uteri from the CC_FLE-immunized mice showed detectable virus, whereas virus was detectable in the uterus of one animal from the WT sE-immunized group.

We have revised the related text describing the comparison between the WT and CC_FLE for **Fig. S8C and S8D**, by deleting the statement “and outperformed WT-sE in this immunocompetent mouse pregnancy protection model”.

Reviewer #2 (Remarks to the Author):

Comment: The authors have addressed most of the comments raised during the previous round of review. The reviewer suggests that the authors cite the references (PMID: 27603093, PMID: 27855206) that firstly report the use of immunocompetent mice in ZIKV research.

Response: We thank the reviewer for the positive feedback and for suggesting references that first reported the use of immunocompetent mice in ZIKV research; these have been added to the text **[Line384]**.

Reviewer #3 (Remarks to the Author):

The authors have satisfactorily addressed all of my concerns.

Response: We thank the reviewer for the positive feedback.

Point-by-Point Response to Reviewer's Comments

Reviewer #1 (Remarks to the Author):

Comment: The authors have addressed most of my comments.

Response: *We thank the reviewer for noting that most comments have been addressed.*

Comment: However, I still believe the data do not fully support the immunogenic superiority of the CC_FLE construct for the following reasons: (a) the 150-loop structure appears to be affected, as seen in the comparison of the JEV construct with the JEV E dimer from the virus (yellow vs. purple);

Response: *We respectfully disagree with the reviewer's statement that the 150-loop structure is affected in the JEV construct based in the comparison between our JEV CC_FLE sE dimer and the sE on the virus (PDB: 5wsn). The 150-loop in both structures are largely identical, with minor variation, which is actually smaller (rmsd 1.19 Å²) than the variation between the JEV sE 150-loop in PDB: 5wsn and PDB: 3p54 (rmsd 1.33 Å²).*

Furthermore, the purpose of the CC_FLE sE construct is not to replicate the sE conformation on the virus particle, but rather to achieve strong immunogenicity while mitigating ADE risk. This rationale is supported by the results from the ZIKV CC_FLE construct. Future immunogenicity studies with the JEV sE CC_FLE will be informative for evaluating the JEV construct.

Comment: (b) the OZ-D4 antibody generated by this antigen binds the dimer in an "unnatural" way that impacts neutralization in authentic virus;

Response: *We acknowledge that the epitope and approach angle of the OZ-D4 antibody are not identical to those of classical EDE-targeting antibodies. However, OZ-D4 represents only one of the monoclonal antibodies elicited by CC_FLE sE. Given the polyclonal serum neutralization specificity presented in this study, which resembles that of EDE1-C8, future work using electron microscopy-based polyclonal epitope mapping (EMPEM) and isolation of additional mAbs from immunized mice will be informative for further characterizing the epitopes and approach angles of EDE-targeting antibodies elicited by CC_FLE sE. We have highlighted this perspective in the revised text. [Lines 552-555]*

Comment: (c) regardless of the viral load in the animal model, the difference between 1/8 and 0/8 is not significant.

Response: *As stated previously, the retro-orbital route directly delivered the virus into the bloodstream, leading to efficient uterine infection and approximately tenfold higher viral loads than in the previous efficacy study. This explains the lack of a statistically significant difference between WT and CC_FLE sE-immunized groups under such high infection levels. Notably, no virus was detected in the uteri of CC_FLE-immunized mice (0/8), whereas one uterus from the WT sE group (1/8) was positive. Although not statistically significant due to the small group size (N = 8), this difference may become clearer with larger cohorts. We have therefore removed the sentence "surpassing the nascent WT sE." [Line 452]*

Comment: I appreciate the effort put into complementing and revising the text according to my suggestions and have no further concerns.

Response: *We appreciate the reviewer's constructive comments, which have indeed led to significant improvements in this manuscript.*